# SCAFF-PD: Communication Efficient Fair and Robust Federated Learning

## ABSTRACT

We present SCAFF-PD, a fast and communication-efficient algorithm for distributionally robust federated learning. Our approach improves fairness by optimizing a family of distributionally robust objectives tailored to heterogeneous clients. We leverage the special structure of these objectives, and design an accelerated primal dual (APD) algorithm which uses bias corrected local steps (as in SCAFFOLD) to achieve significant gains in communication efficiency and convergence speed. We evaluate SCAFF-PD on several benchmark datasets and demonstrate its effectiveness in improving fairness and robustness while maintaining competitive accuracy. Our results suggest that SCAFF-PD is a promising approach for federated learning in resource-constrained and heterogeneous settings.

## 1 INTRODUCTION

Federated learning is a popular approach for training machine learning models on decentralized data, where data privacy concerns or other constraints prevent centralized data aggregation (McMahan et al., 2017; Kairouz et al., 2021). In federated learning, model updates are computed locally on each device (the *client*) and then aggregated to train a global model at the center (the *server*). This approach has gained traction due to its ability to leverage data from multiple sources while preserving privacy, security, and autonomy, and has the potential to make machine learning more participatory in a range of interesting problem domains (Kulynych et al., 2020; Jones & Tonetti, 2020; Pentland et al., 2021).

Federated learning is naturally most attractive when the participating clients have access to different data, leading to data heterogeneity (du Terrail et al., 2022). This heterogeneity can lead to significant fairness issues, where the performance of the global model can be biased towards the data distribution of some clients over others (Dwork et al., 2012; Li et al., 2019; Abay et al., 2020). Heterogeneity can also hurt the generalization of the global model (Quinonero-Candela et al., 2008; Mohri et al., 2019). Specifically, if some clients have a disproportionate influence on the global model, the resulting model is neither fair nor will it generalize well to new clients. Such disparities are especially prevalent and detrimental in medical research, and have resulted in misdiagnosis and suboptimal treatment (Graham, 2015; Albain et al., 2009; Nana-Sinkam et al., 2021).

To address these challenges, distributionally robust objectives (DRO) explicitly account for the heterogeneity across clients and seek to optimize performance under the worst-case data distribution across clients, rather than just the average performance (Rahimian & Mehrotra, 2019). This approach can lead to more robust models that are less biased towards specific clients and more likely to generalize to new clients (Mohri et al., 2019; Duchi et al., 2023). However, such robust objectives are significantly harder to optimize. Current algorithms have very slow convergence, potentially to the point of being impractical (Ro et al., 2021). This leads to the central question of our work:

*Can we design federated optimization techniques for the DRO problem with convergence rates that match their average objective counterparts?*

### 1.1 OUR CONTRIBUTIONS.

We summarize our contributions below.

**(1) Framework.** We present a general formulation for the cross-silo federated DRO problem:

$$\min_{\boldsymbol{x}} \max_{\boldsymbol{\lambda} \in \Lambda} \left\{ F(\boldsymbol{x}, \boldsymbol{\lambda}) := \sum_{i=1}^{N} \lambda_i \cdot f_i(\boldsymbol{x}) - \psi(\boldsymbol{\lambda}) \right\}, \tag{1.1}$$

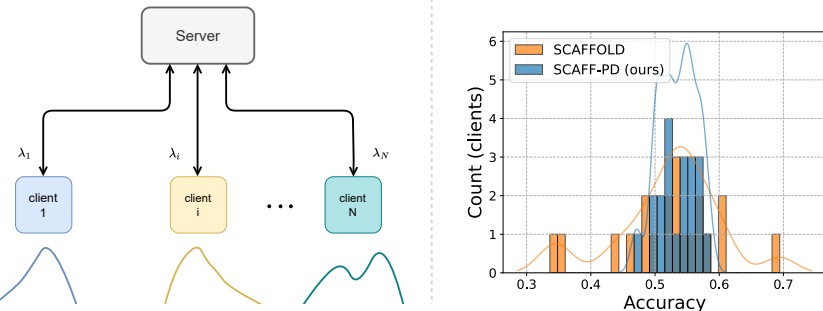

Figure 1: (**left**) In federated learning, the data distribution across individual clients differ significantly from one another. (**right**) When directly applying SOTA federated optimization algorithm (SCAFFOLD), the learned global model is biased toward certain clients, leading to noticeably worse performance when applied to a subset of participating clients. Our proposed algorithm—SCAFF-PD—largely mitigates this bias via learning a distributionally robust global model, which significantly enhances the performance of the most challenging subset of clients, specifically the worst 20%.

where $f_i(\boldsymbol{x})$ is the loss suffered by client $i$. Instead of minimizing a simple average of the client losses, equation (1.1) incorporates weights using $\boldsymbol{\lambda} \in \mathbb{R}^N$. The choice of $\boldsymbol{\lambda}$ is made in a *worst-case* manner, while being subject to the constraint set $\Lambda$ and regularized with $\psi(\boldsymbol{\lambda})$. As we will show, this formulation is a generalization of several specific fair objectives that have been proposed in the federated learning literature (Mohri et al., 2019; Li et al., 2019; 2020a; Zhang et al., 2022a; Pillutla et al., 2021).

**Algorithm.** The objective defined in equation (1.1) is a min-max problem and can be directly optimized using well-established algorithms such as gradient descent ascent (GDA). However, such approaches ignore the unique structure of our formulation, particularly the linearity of the interaction term between $\boldsymbol{\lambda}$ and $\boldsymbol{x}$. We leverage this to design an accelerated primal-dual (APD) algorithm (Hamedani & Aybat, 2021). Additionally, we propose to use control variates (à la SCAFFOLD) to correct the bias caused by local steps, making optimal use of local client computation (Karimireddy et al., 2020). Our proposed method, SCAFF-PD, combines these ideas to provide an efficient and practical algorithm, compatible with secure aggregation.

**Convergence.** We provide strong convergence guarantees for SCAFF-PD when $f_i$ are strongly convex. If $\psi(\boldsymbol{\lambda})$ is a generic convex function, we achieve an accelerated $O\left(1/R^2\right)$ rate of convergence. Furthermore, if $\psi$ is strongly convex, SCAFF-PD converges linearly at a rate of $\exp\left(-O(R)\right)$. This represents the first federated approach for the DRO problem that achieves *linear* convergence, let alone an *accelerated* rate. Finally, we extend our analysis to the stochastic setting, where we obtain an optimal rate of $O\left(1/R\right)$, and improve over the previous $O(1/\sqrt{R})$ rate. Thus, we show that the sample complexity as well as the communication complexity for the DRO problem matches that of the easier average objective.

**Practical Evaluation.** We conducted comprehensive simulations and demonstrate accelerated convergence, robustness to data heterogeneity, and the ability to leverage local computations.

For deep learning models, we avail ourselves of a two-stage Train-Convexify-Train method (Yu et al., 2022). First, we train a deep learning model using conventional federated learning methods, such as FedAvg. Then, we apply SCAFF-PD to fine tune a convex approximation. To evaluate our algorithms, we use several real-world datasets with various distributionally robust objectives, and we study the trade-off between the mean and tail accuracy of these methods.

## 2 RELATED WORK

**Cross-silo FL.** Federated learning (FL) is a distributed machine learning paradigm that enables model training without exchanging raw data. In cross-silo FL (which is our focus), valuable data is split across different organizations, and each organization is either protected by privacy regulations or unwilling to share their raw data. Such organizations are referred to as "data islands" and can be found in hospital networks, financial institutions, autonomous-vehicle companies, etc. Thus, cross-silo FL involves a few highly reliable clients who potentially have extremely diverse data.

The most widely used federated optimization algorithm is Federated Averaging (FedAvg) (McMahan et al., 2017), which averages the local model updates to produce a global model. However, FedAvg is known to suffer from poor convergence when the local datasets are heterogeneous (Hsieh et al., 2020; Li et al., 2020b; Karimireddy et al., 2020; Reddi et al., 2021; Wang et al., 2021; du Terrail et al., 2022, etc.). Scaffold (Karimireddy et al., 2020) corrects for this heterogeneity, leading to more accurate updates and faster convergence (Mishchenko et al., 2022; Li et al., 2022a; Yu et al., 2022). However, all of these methods are restricted to optimizing the average of the client objectives.

**Distributionally Robust Optimization.** DRO is a framework for optimization under uncertainty, where the goal is to optimize the worst-case performance over a set of probability distributions. See Rahimian & Mehrotra (2019) for a review and its history in risk management, economics, and finance. Fast centralized optimization methods have been developed when uncertainity is represented by $f$-divergences (Wiesemann et al., 2014; Namkoong & Duchi, 2016; Levy et al., 2020) or Wasserstein distances (Mohajerin Esfahani & Kuhn, 2018; Gao & Kleywegt, 2022). The former approach accounts for changing proportions of subpopulations, relating it to notions of subpopulation fairness (Duchi et al., 2023; Santurkar et al., 2020; Piratla et al., 2021; Martinez et al., 2021). Our work also implicitly focuses on $f$-divergences. Deng et al. (2020) and Zecchin et al. (2022) adapt the gradient-descent-ascent (GDA) algorithm to solve the federated and decentralized DRO problems respectively. However, these methods inherit the slowness of both the GDA and FedAvg algorithms, making their performance trail the state of the art for the average objective (Mishchenko et al., 2022).

**Fairness in FL.** While fairness is an extremely multi-faceted concept, here we are concerned with the distribution of model performance across clients. Mohri et al. (2019) noted that minimizing the average of the client losses may lead to unfair distribution of errors, and instead proposed an *agnostic FL* (AFL) framework which minimizes a worst-case mixture of the client losses. Alternatives and extensions to AFL have also been proposed subsequently Li et al. (2019; 2020a); Pillutla et al. (2021). Again, the convergence of optimization methods for these losses (when analyzed) is significantly slower than their centralized counterparts.

While all of these works demand equitable performance across all clients, others propose to scale a client's accuracy in proportion to their contribution (Sim et al., 2020; Blum et al., 2021; Xu et al., 2021; Zhang et al., 2022a; Karimireddy et al., 2022). These methods are motivated by game-theoretic considerations to incentivize clients and improve the quality of the data contributions. Our framework (1.1) can be applied to such mechanisms by an appropriate choice of $\{f_i\}, \Lambda$, and $\psi$. For example, Zhang et al. (2022a) show how to set these to recover the Nash bargaining solution (Nash Jr, 1950). Thus, our work can be seen as a practical optimization algorithm to implement many of the mechanisms studied in FL.

Finally, personalization—serving a separate model to each client—has also been proposed as a method to improve the distribution of client performance (Yu et al., 2020). However, personalized models are sometimes not feasible either due to regulations (Vokinger et al., 2021) or because the client may not have additional data. Further, personalization does not remove the differences in performance (though it does reduce it) (Yu et al., 2020), nor does it solve the game-theoretic considerations described above. Extending our work to this setting is an important question we leave for future work.

## 3 PROBLEM SETUP

We consider the min-max optimization problem in the context of federated learning, where the objective function, defined in Eq. (1.1), is distributed among $N$ clients. Each $f_i : \mathbb{R}^d \to \mathbb{R}$ is the local function on the $i$-th client, where $f_i(\boldsymbol{x}) = \mathbb{E}_{\xi \sim \mathcal{D}_i}[f(\boldsymbol{x}, \xi)]$ and $\mathcal{D}_i$ is the data distribution of the $i$-th client. For example, we can define $\mathcal{D}_i$ as the uniform distribution over the training dataset present on the $i$-th client.

**Notation.** We use the notation $\boldsymbol{x}^r \in \mathbb{R}^d$ to denote the global iterate at the $r$-th round, and use $\boldsymbol{u}_{i,j}^r \in \mathbb{R}^d$ to denote the local iterate at the $j$-th step on the $i$-th client (at the $r$-th round). We apply $\boldsymbol{\lambda} = [\lambda_1, \dots, \lambda_N]^\top \in \mathbb{R}^N$ to denote the weight vector, where $\lambda_i$ is the weight for client $i$, and we let $\mathrm{D}(\boldsymbol{\lambda}, \boldsymbol{\lambda}') = \|\boldsymbol{\lambda} - \boldsymbol{\lambda}'\|^2/2$. We let $[N]$ denote the set $\{1, \dots, N\}$. To facilitate clarity and presentation, we let

$$\Phi(\boldsymbol{x}, \boldsymbol{\lambda}) = \sum_{i=1}^{N} \lambda_i \cdot f_i(\boldsymbol{x}). \tag{3.1}$$

For local gradients, we let $g_i(\boldsymbol{u}_{i,j-1})$ denote the stochastic gradient of $f_i$ at iterate $\boldsymbol{u}_{i,j-1}$:

$$g_i(\boldsymbol{u}_{i,j-1}^r) = \nabla f_i(\boldsymbol{u}_{i,j-1}^r, \xi_{i,j-1}^r). \tag{3.2}$$

**Choosing $\psi$ and $\Lambda$.** We let $\psi : \mathbb{R}^N \to \mathbb{R}$ denote the regularization on the weight vector $\boldsymbol{\lambda}$. The $\chi^2$ penalty (Levy et al., 2020) involves setting

$$\psi(\boldsymbol{\lambda}) = \mathrm{D}_{\chi^2}(\boldsymbol{\lambda}) = \frac{\rho}{2N} \sum_{i=1}^{N} (N\lambda_i - 1)^2, \text{ and } \Lambda = \Delta^N. \tag{3.3}$$

When regularization is set to zero with $\rho = 0$, the DRO formulation (1.1) recovers the agnostic federated learning (AFL) of Mohri et al. (2019). A non-zero value of $\rho$ can be used to trade off the worst-case loss against the average loss. In particular, setting $\rho \to \infty$ recovers the standard average FL objective. While we will primarily focus on (3.3) in this work, other choices are also possible. The DRO objective becomes the $\alpha$-Conditional Value at Risk (CVaR) loss (Duchi & Namkoong, 2021), also known as super-quantile loss (Pillutla et al., 2021) by setting

$$\psi(\boldsymbol{\lambda}) = 0, \text{ and } \Lambda = \{\boldsymbol{\lambda} \in \Delta, \lambda_i \leq 1/(\alpha N)\}.$$

Finally, we can recover the Q-FL loss of Li et al. (2019) by setting

$$\psi(\boldsymbol{\lambda}) = \|\boldsymbol{\lambda}\|^{1+\frac{1}{q}}, \text{ and } \Lambda = \mathbb{R}^N.$$

**Definitions and assumptions.** In the convergence analysis of our proposed algorithms, we rely on the following definitions and assumptions regarding the local functions and the regularization term $\psi$:

**Definition 3.1 (Smoothness)** $f(\cdot)$ *is convex and differentiable, and there exists $L \geq 0$ such that for any $\boldsymbol{x}_1, \boldsymbol{x}_2$ in the domain of $f_i(\cdot)$,*

$$\|\nabla f_i(\boldsymbol{x}_1) - \nabla f_i(\boldsymbol{x}_2)\| \leq L\|\boldsymbol{x}_1 - \boldsymbol{x}_2\|. \tag{3.4}$$

**Definition 3.2 (Strong convexity)** $f(\cdot)$ *is $\mu$-strongly convex, i.e.,*

$$f(\boldsymbol{x}_2) \geq f(\boldsymbol{x}_1) + \langle \nabla f(\boldsymbol{x}_1), \boldsymbol{x}_2 - \boldsymbol{x}_1 \rangle + \frac{\mu}{2}\|\boldsymbol{x}_2 - \boldsymbol{x}_1\|^2. \tag{3.5}$$

**Assumption 3.3 (Smoothness w.r.t. $\Phi$)** $\Phi(\boldsymbol{x}, \cdot)$ *is concave and differentiable, and there exists $L_{\boldsymbol{\lambda x}} \geq 0$ such that for any $\boldsymbol{x}_1, \boldsymbol{x}_2$ in the domain of $\Phi(\cdot, \boldsymbol{\lambda})$ and $\boldsymbol{\lambda}_1, \boldsymbol{\lambda}_2$ in the domain of $\Phi(\boldsymbol{x}, \cdot)$,*

$$\|\nabla_{\boldsymbol{\lambda}}\Phi(\boldsymbol{x}_1, \boldsymbol{\lambda}_1) - \nabla_{\boldsymbol{\lambda}}\Phi(\boldsymbol{x}_2, \boldsymbol{\lambda}_2)\| \leq L_{\boldsymbol{\lambda x}}\|\boldsymbol{x}_1 - \boldsymbol{x}_2\|. \tag{3.6}$$

**Assumption 3.4 (Bounded noise)** *There exist $\zeta \geq 0$ such that for all $i \in [N]$, the local gradient $g_i(\boldsymbol{x})$ defined in Eq. (3.2) satisfies*

$$\mathbb{E}\left[\|g_i(\boldsymbol{x}) - \nabla f_i(\boldsymbol{x})\|^2\right] \leq \zeta^2, \quad \mathbb{E}[g_i(\boldsymbol{x})] = \nabla f_i(\boldsymbol{x}). \tag{3.7}$$

## 4 SCAFF-PD: ACCELERATED PRIMAL-DUAL FEDERATED ALGORITHM WITH BIAS CORRECTED LOCAL STEPS

In this section, we describe our proposed algorithm SCAFF-PD (Stochastic Controlled Averaging with Primal-Dual updates) for solving the federated DRO problem (1.1). We present the pseudo-code for SCAFF-PD in Algorithm 1 and algorithm used for local updates in Algorithm 2.

As described in Algorithm 1, SCAFF-PD comprises three main steps that are executed at each communication round $r$: (1). Collecting loss vector $[L_1^r, \dots, L_N^r]^\top$ and gradients $\{g_i(\boldsymbol{x}^r)\}_{i=1}^N$ (for bias correction); (2). Update to the dual variable by Eq. (4.1); (3). Local updates to each client model, and aggregating the updates by using the updated dual variable, i.e., Eq. (4.1). We provide the pseudo-code for local updates in Algorithm 2.

**Extrapolated Dual Update.** Based on the computed loss vector $\nabla_{\boldsymbol{\lambda}}\Phi(\boldsymbol{x}^r, \boldsymbol{\lambda}^r) = [L_1^r, \dots, L_N^r]^\top$ in the first step, we update the weight vector $\boldsymbol{\lambda}$. Importantly, when $\theta_r > 0$, we use both the dual gradient from the current round ($\nabla_{\boldsymbol{\lambda}}\Phi(\boldsymbol{x}^r, \boldsymbol{\lambda}^r)$) as well as the past round ($\nabla_{\boldsymbol{\lambda}}\Phi(\boldsymbol{x}^{r-1}, \boldsymbol{\lambda}^{r-1})$) to

---

**Algorithm 1** SCAFF-PD($\boldsymbol{x}^0, \boldsymbol{\lambda}^0$)

---

**for** $r = 1, 2, \ldots, R$ **do**
   # (1).Collect gradient and loss vector
   Set parameters $\{\tau_r, \sigma_r, \gamma_r, \theta_r\}$
   **for** $i = 1, 2, \ldots, N$ **do**
     $L_i^r = f_i(\boldsymbol{x}^r)$, $\boldsymbol{c}_i^r = g_i(\boldsymbol{x}^r)$, Communicate $(L_i^r, \boldsymbol{c}_i^r)$ to center
   # (2).Update dual $\boldsymbol{\lambda}$

$$\boldsymbol{s}^r = (1 + \theta_r)\nabla_{\boldsymbol{\lambda}}\Phi(\boldsymbol{x}^r, \boldsymbol{\lambda}^r) - \theta_r\nabla_{\boldsymbol{\lambda}}\Phi(\boldsymbol{x}^{r-1}, \boldsymbol{\lambda}^{r-1})$$

$$\boldsymbol{\lambda}^{r+1} = \text{argmin}_{\boldsymbol{\lambda} \in \Lambda}\left\{\psi(\boldsymbol{\lambda}) - \langle\boldsymbol{s}^r, \boldsymbol{\lambda}\rangle + \frac{1}{\sigma_r}\text{D}(\boldsymbol{\lambda}, \boldsymbol{\lambda}^r)\right\} \tag{4.1}$$

   # (3).Update primal $\boldsymbol{x}$
   $\boldsymbol{c}^r = \sum_{i=1}^{N}\lambda_i^{r+1}\boldsymbol{c}_i^r$, Communicate $\boldsymbol{c}^r$ to each client
   **for** $i = 1, 2, \ldots, N$ **do**
     $\Delta\boldsymbol{u}_i^r \leftarrow$ LOCAL-UPDATE$(\boldsymbol{x}^r, \boldsymbol{c}_i^r, \boldsymbol{c}^r)$, Communicate $\Delta\boldsymbol{u}_i^r$ to the center
   Aggregate updates from different client via the weight vector $\boldsymbol{\lambda}^{r+1}$

$$\boldsymbol{x}^{r+1} = \text{argmin}_{\boldsymbol{x}}\left\{\langle\sum_{i=1}^{N}\lambda_i^{r+1}\Delta\boldsymbol{u}_i^r, \boldsymbol{x}\rangle + \frac{1}{\tau_r}\text{D}(\boldsymbol{x}, \boldsymbol{x}^r)\right\} \tag{4.2}$$

**Return:** $(\boldsymbol{x}^{R+1}, \boldsymbol{\lambda}^{R+1})$

---

**Algorithm 2** LOCAL-UPDATE$(\boldsymbol{x}, \boldsymbol{c}_i, \boldsymbol{c})$

---

**Input:** optimization parameters $(\eta_\ell, J)$, model parameters $(\boldsymbol{c}_i, \boldsymbol{c}, \boldsymbol{x})$
$\boldsymbol{u}_{i,0} = \boldsymbol{x}$
**for** $j = 1, 2, \ldots, J$ **do**
   $\boldsymbol{u}_{i,j} = \boldsymbol{u}_{i,j-1} - \eta_\ell \cdot (g_i(\boldsymbol{u}_{i,j-1}) - \boldsymbol{c}_i + \boldsymbol{c})$
$\Delta\boldsymbol{u}_i = (\boldsymbol{x} - \boldsymbol{u}_{i,J})/(\eta_\ell J)$
**Return:** $\Delta\boldsymbol{u}_i$

---

obtain the extrapolated gradient $\boldsymbol{s}^r$. The gradient extrapolation step is widely used in primal-dual hybrid gradient (PDHG) methods (Chambolle & Pock, 2016) for solving convex-concave saddle-point problems, and it provides the key component in our algorithm for achieving acceleration. The extrapolation step used in Eq. (4.1) is to Nesterov's acceleration (Nesterov, 2003), which can lead to faster convergence rate and has been widely utilized for achieving acceleration in solving various optimization problems. (Chambolle & Pock, 2011; 2016; Zhang & Lin, 2015; Hamedani & Aybat, 2021).

**Local Steps and Control Variates $\boldsymbol{c}_i$.** Supposing that communication is not a limiting factor, each client can compute its local gradient and transmit it to the server without any local steps. In this case, the update to the primal variable $\boldsymbol{x}$ becomes

$$\Delta\boldsymbol{u}_i^r = g_i(\boldsymbol{x}^r), \quad \text{argmin}_{\boldsymbol{x}}\left\{\langle\sum_{i=1}^{N}\lambda_i^{r+1}g_i(\boldsymbol{x}^r), \boldsymbol{x}\rangle + \frac{1}{\tau_r}\text{D}(\boldsymbol{x}, \boldsymbol{x}^r)\right\}. \tag{4.3}$$

This update performs the primal update with the unbiased gradient $\nabla_{\boldsymbol{x}}F(\boldsymbol{x}^r, \boldsymbol{\lambda}^{r+1})$, which is equivalent to the standard primal update in primal-dual-based algorithms (Chambolle & Pock, 2016; Hamedani & Aybat, 2021; Zhang et al., 2022b). However, such an update does not effectively utilize the local computational resources available on each client. Hence, we would like to perform multiple local update steps. The catch is that performing multiple local steps is known to lead to biased updates and "client-drift" (Karimireddy et al., 2020; Woodworth et al., 2020; Wang et al., 2020). We explicitly correct for this bias using control variates $\{\boldsymbol{c}_i\}_{i\in[N]}$ similar to SCAFFOLD. As we will demonstrate in the subsequent theoretical analysis, this correction allows SCAFF-PD to converge to the saddle-point solution of the DRO problem regardless of the data heterogeneity.

While we use local updates on the primal variable, we do not perform any on the dual variable. This is unlike general federated min-max optimization algorithms (Hou et al., 2021; Beznosikov et al.,

2022). This design aligns well with the federated DRO formulation since it is impractical for each client to update the weight vector at each local step due to their lack of knowledge regarding the loss values of other clients. The aggregation of SCAFF-PD on the server resembles federated algorithms used for solving minimization problems, with the key difference being the utilization of the updated weight vector for primal aggregation.

## 5 THEORETICAL ANALYSIS

We now present the convergence results for SCAFF-PD in solving the min-max optimization problem described in Eq. (1.1). Firstly, in Section 5.1, we introduce the results for the strongly-convex-concave setting. Subsequently, in Section 5.2, we present the results for the strongly-convex-convex setting.

### 5.1 STRONGLY-CONVEX-CONCAVE SETTING

We first introduce how to choice the parameters for SCAFF-PD in when $\psi$ is convex and $\{f_i\}_{i \in [N]}$ are strongly convex in Condition 5.1.

**Condition 5.1** *The parameters of Algorithm 1 are defined as*

$$\sigma_{-1} = \gamma_0 \bar{\tau}, \quad \sigma_r = \gamma_r \tau_r, \quad \theta_r = \sigma_{r-1}/\sigma_r, \quad \gamma_{r+1} = \gamma_r(1 + \mu_{\boldsymbol{x}} \tau_r). \tag{5.1}$$

Next we present our convergence results in this setting.

**Theorem 5.1** *Supppose $\{f_i\}_{i \in [N]}$ are $\mu_{\boldsymbol{x}}$-strongly convex and $L_{\boldsymbol{xx}}$-smooth. If Assumption 3.3 and Assumption 3.4 hold, and we let the parameters $\{\tau_r, \sigma_r, \gamma_r, \theta_r\}$ of Algorithm 1 satisfy Condition 5.1, then the R-th iterate $(\boldsymbol{x}^R, \boldsymbol{\lambda}^R)$ satisfies*

$$\mathbb{E}\left[\|\boldsymbol{x}^R - \boldsymbol{x}^\star\|^2\right] \leq \frac{C_1}{R^2}\left[\|\boldsymbol{x}^\star - \boldsymbol{x}^0\|^2 + \|\boldsymbol{\lambda}^0 - \boldsymbol{\lambda}^\star\|^2\right] + \frac{C_2}{R}\zeta^2, \tag{5.2}$$

*where $C_1, C_2 \geq 0$ are non-negative constants.*

**Corollary 5.2** *Under the assumptions in Theorem 5.1,*

- *(deterministic local gradient): If the local gradient satisfies $g_i(\boldsymbol{x}) = \nabla f_i(\boldsymbol{x})$ for $i \in [N]$, then after $O\left(\frac{\|\boldsymbol{x}^\star - \boldsymbol{x}^0\|^2 + \|\boldsymbol{\lambda}^0 - \boldsymbol{\lambda}^\star\|^2}{\sqrt{\varepsilon}}\right)$ rounds, we have $\|\boldsymbol{x}^R - \boldsymbol{x}^\star\|^2 \leq \varepsilon$.*

- *(stochastic local gradient): If the local gradient satisfies Assumption 3.4 with $\sigma > 0$, then after $O\left(\frac{\|\boldsymbol{x}^\star - \boldsymbol{x}^0\|^2 + \|\boldsymbol{\lambda}^0 - \boldsymbol{\lambda}^\star\|^2}{\sqrt{\varepsilon}} + \frac{\zeta^2}{\varepsilon}\right)$ rounds, we have $\mathbb{E}\left[\|\boldsymbol{x}^R - \boldsymbol{x}^\star\|^2\right] \leq \varepsilon$.*

**Remark 5.3** *As suggested by the Corollary 5.2, in the deterministic setting ($\zeta = 0$, when applying SCAFF-PD for solving the min-max problems in the vanilla AFL and the super-quantile approach, SCAFF-PD achieves the convergence rate of $O(1/R^2)$. The rate of SCAFF-PD is faster than existing algorithms – the convergence rate is $O(1/R)$ in both Mohri et al. (2019); Pillutla et al. (2021). In addition, the algorithm with theoretical convergence guarantees introduced in Mohri et al. (2019) does not apply local steps (i.e., number of local updates $J = 1$), resulting in inferior performance in practical applications.*

**Remark 5.4** SCAFF-PD *matches the rates ($O(1/R^2)$) of the centralized accelerated primal-dual algorithm (Hamedani & Aybat, 2021) when $\zeta = 0$. Meanwhile, our proposed algorithm converges faster compared to directly applying centralized gradient descent ascent (GDA) and extra-gradient method (EG) for solving Eq. (1.1), which achieve a rate of $O(1/R)$.*

### 5.2 STRONGLY-CONVEX-STRONGLY-CONCAVE SETTING

We next present results for the strongly-convex-strongly-concave setting. Differing from the strongly-convex-concave setting, the parameters of Algorithm 1 are fixed across different rounds, as follows.

**Condition 5.2** *The parameters of Algorithm 1 are defined as*

$$\mu_{\boldsymbol{x}}\tau = O\left(\frac{1-\theta}{\theta}\right), \quad \mu_{\boldsymbol{\lambda}}\sigma = O\left(\frac{1-\theta}{\theta}\right), \quad \frac{1}{1-\theta} = O\left(\left(\frac{L_{\boldsymbol{xx}}}{\mu_{\boldsymbol{x}}} + \sqrt{\frac{L_{\boldsymbol{\lambda x}}^2}{\mu_{\boldsymbol{x}}\mu_{\boldsymbol{\lambda}}}}\right) \vee \frac{\zeta^2}{\mu_{\boldsymbol{x}}\varepsilon}\right). \tag{5.3}$$

**Theorem 5.5** *Suppose $\{f_i\}_{i\in[N]}$ are $\mu_{\boldsymbol{x}}$-strongly convex and $L_{\boldsymbol{xx}}$-smooth, and $\psi$ is $\mu_{\boldsymbol{y}}$-strongly convex. If Assumption 3.3 and Assumption 3.4 hold, and we let the parameters $\{\tau, \sigma, \theta\}$ of Algorithm 1 satisfy Condition 5.2, then the $R$-th iterate $(\boldsymbol{x}^R, \boldsymbol{\lambda}^R)$ satisfies*

$$\mathbb{E}\left[\mu_{\boldsymbol{x}}\|\boldsymbol{x}^r - \boldsymbol{x}^\star\|^2\right] \leq C_1 \theta^R \left[\|\boldsymbol{x}^0 - \boldsymbol{x}^\star\|^2 + \|\boldsymbol{\lambda}^0 - \boldsymbol{\lambda}^\star\|^2\right] + C_2(1-\theta)\frac{\zeta^2}{\mu_{\boldsymbol{x}}}, \qquad (5.4)$$

*where $C_1, C_2 \geq 0$ are non-negative constants.*

**Corollary 5.6** *Under the assumptions in Theorem 5.5,*

- *(deterministic local gradient): If the local gradient satisfies $g_i(\boldsymbol{x}) = \nabla f_i(\boldsymbol{x})$ for $i \in [N]$, then after $O\left(\left(\frac{L_{\boldsymbol{xx}}}{\mu_{\boldsymbol{x}}} + \sqrt{\frac{L_{\boldsymbol{\lambda x}}^2}{\mu_{\boldsymbol{x}}\mu_{\boldsymbol{\lambda}}}}\right)\log\left(\frac{\|\boldsymbol{x}^0-\boldsymbol{x}^\star\|^2 + \|\boldsymbol{\lambda}^0-\boldsymbol{\lambda}^\star\|^2}{\varepsilon}\right)\right)$ rounds, $\mu_{\boldsymbol{x}}\|\boldsymbol{x}^R - \boldsymbol{x}^\star\|^2 \leq \varepsilon$.*

- *(stochastic local gradient): If the local gradient satisfies Assumption 3.4 with $\zeta > 0$, then after $O\left(\left(\frac{L_{\boldsymbol{xx}}}{\mu_{\boldsymbol{x}}} + \sqrt{\frac{L_{\boldsymbol{\lambda x}}^2}{\mu_{\boldsymbol{x}}\mu_{\boldsymbol{\lambda}}}} + \frac{\zeta^2}{\mu_{\boldsymbol{x}}\varepsilon}\right)\log\left(\frac{\|\boldsymbol{x}^0-\boldsymbol{x}^\star\|^2 + \|\boldsymbol{\lambda}^0-\boldsymbol{\lambda}^\star\|^2}{\varepsilon}\right)\right)$ rounds, $\mathbb{E}\left[\mu_{\boldsymbol{x}}\|\boldsymbol{x}^R - \boldsymbol{x}^\star\|^2\right] \leq \varepsilon$.*

**Remark 5.7** *Our algorithm converges linearly to the global saddle point when each client applies a noiseless gradient for local updates (i.e., $\zeta = 0$) in the presence of data heterogeneity and client-drift in federated learning. In contrast, previous approaches exhibit only sub-linear convergence. In the strongly-convex-strongly-concave setting, DRFA (Deng et al., 2020) converges to the saddle-point solution with rate $O(1/R)$ when there is no data heterogeneity and $\zeta = 0$.*

**Remark 5.8** *By applying bias correction in local updates, the convergence rates of our algorithm match those of the centralized accelerated primal-dual algorithm (Zhang et al., 2021) in both deterministic and stochastic settings.*

**Remark 5.9** *Compared to the standard minimization in federated learning, the DRO objective results in a slightly worse condition number in terms of convergence rate. In comparison to the standard minimization objective in federated learning, the DRO objective yields a slightly worse condition number. Solving DRO with SCAFF-PD requires $(\sqrt{L_{\boldsymbol{xx}}/\mu_{\boldsymbol{x}}} + \sqrt{L_{\boldsymbol{\lambda x}}^2/(L_{\boldsymbol{xx}}\mu_{\boldsymbol{\lambda}})})$ times more communication rounds compared to solving minimization problems with ProxSkip (Mishchenko et al., 2022).*

## 6 EXPERIMENTS

We now study the performance of SCAFF-PD for solving federated DRO problems on both synthetic datasets and real-world datasets. Our primary objective when working with synthetic datasets is to validate the convergence analysis of SCAFF-PD. On real-world datasets, we compare with existing federated optimization algorithms for learning robust and fair models (DRFA (Deng et al., 2020), AFL (Mohri et al., 2019), and $q$-FFL (Li et al., 2019)) as well as widely used federated algorithms for solving minimization problems including FedAvg (McMahan et al., 2017) and SCAFFOLD (Karimireddy et al., 2020). After conducting thorough evaluations, we have observed that our proposed accelerated algorithms achieve fast convergence rates and strong empirical performance on real-world datasets. We have provided supplementary experimental results in Appendix C, which includes additional baseline methods, ablations on our algorithm, and other relevant findings.

### 6.1 RESULTS ON SYNTHETIC DATASETS

To construct the synthetic datasets, we follow the setup described in Eq. (1.1) and consider a simple robust regression problem. Specifically, for the $i$-th client, the local function $f_i$ is defined as $f_i(\boldsymbol{x}) = \frac{1}{m_i}\sum_{j=1}^{m_i}(\langle \boldsymbol{a}_i^j, \boldsymbol{x}\rangle - y_i^j)^2 + \frac{\mu_{\boldsymbol{x}}}{2}\|\boldsymbol{x}\|^2$, where $j$ is sample index on this client and there are $m_i$ training samples on client-$i$. We apply the $\chi^2$ penalty for regularizing the weight vector $\boldsymbol{\lambda}$. To generate the data, each input $\boldsymbol{a}_i^j$ is sampled from a Gaussian distribution $\boldsymbol{a}_i^j \sim \mathcal{N}(\boldsymbol{0}, \boldsymbol{I}_{d\times d})$. Then we random generate $\widehat{\boldsymbol{x}} \sim \mathcal{N}(\boldsymbol{0}, c^2\boldsymbol{I}_{d\times d})$, and $\delta_i^{\boldsymbol{x}} \sim \mathcal{N}(\boldsymbol{0}, \sigma^2\boldsymbol{I}_{d\times d})$. Based on $(\widehat{\boldsymbol{x}}, \delta_i^{\boldsymbol{x}})$, we generate $y_i^i$ as $y_i^i = \langle \boldsymbol{a}_i^j, \widehat{\boldsymbol{x}} + \delta_i^{\boldsymbol{x}}\rangle$. Therefore, there exist distribution shifts across different clients (i.e., concept

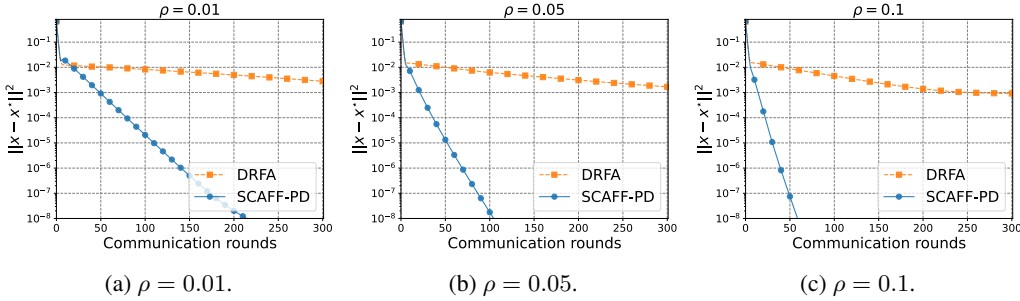

(a) $\rho = 0.01$.        (b) $\rho = 0.05$.        (c) $\rho = 0.1$.

Figure 2: We compare our proposed algorithm with the existing method DFRA (Deng et al., 2020) on synthetic datasets. $\rho$ is the strength of regularization $\psi$ (defined in Eq. (3.3)). $X$-axis represents the number of communication rounds, and $Y$-axis represents the distance to optimal solution.

shifts). We set $N = 5$, $d = 10$, and $m_i = 100$ for $i \in [N]$. To measure the algorithm performance, we evaluate the distance between $\boldsymbol{x}^R$ and the optimal solution $\boldsymbol{x}^\star$: $\|\boldsymbol{x}^R - \boldsymbol{x}^\star\|^2$.

We compare SCAFF-PD with DRFA (Deng et al., 2020) on this synthetic dataset. The regularization parameter $\rho$ for $\psi$ is varied from $0.01$ to $0.1$. For both algorithms, we set the number of local steps to be $100$ and select the algorithm parameters through grid search. The comparison results are summarized in Fig 2. As shown in Fig 2, we observe that our proposed algorithm SCAFF-PD achieves linear convergence rates in all three settings. In contrast, DRFA converges much more slowly compared to SCAFF-PD. We have included more experimental results under this synthetic setup in Appendix C, including results on the effect of local steps and data heterogeneity.

## 6.2 RESULTS ON REAL-WORLD DATASETS

**Dataset setup.** We evaluate the performance of various federated learning algorithms on CIFAR100 (Krizhevsky et al., 2009) and TinyImageNet (Le & Yang, 2015). We follow the setup used in Li et al. (2022b): we consider different degrees of data heterogeneity by applying Dirichlet allocation, denoted by $\text{Dir}(\alpha)$, to partition the dataset into different clients. Smaller $\alpha$ values in $\text{Dir}(\alpha)$ leads to higher data heterogeneity. Additionally, after the data partition through the Dirichlet allocation, we randomly sample $30\%$ of the clients and remove $70\%$ training samples from those clients. Such a sub-sampling procedure can better model real-world data-imbalance scenarios. We consider the number of clients $N = 20$ for both datasets. Results on larger number of clients and other real-world datasets can be found in Appendix C.

**Model setup.** We consider learning a linear classifier by using representations extracted from pretrained deep neural networks. Previous studies have demonstrated the efficacy of this approach, particularly in the context of data heterogeneity (Yu et al., 2022) as well as sub-group robustness (Izmailov et al., 2022). For both datasets, we apply the ResNet-18 (He et al., 2016) pre-trained on ImageNet-1k (Deng et al., 2009) as the backbone for extracting feature representations of the image samples. To apply the pre-trained ResNet-18, we resize the images from CIFAR100 and TinyImageNet to $3 \times 224 \times 224$.

**Comparisons with existing approaches.** We consider three data heterogeneity settings for both datasets. To measure the performance of different algorithms, besides the average classification accuracy across clients, we also evaluate the `worst-20%` accuracy[1] for comparing fairness and robustness of different federated learning algorithms. Previous studies have employed this metric for comparing different model in federated learning Li et al. (2019). The comparative results are summarized in Table 1. We find that our proposed algorithm outperforms existing methods in most settings, especially under higher heterogeneity. For example, when the level of data heterogeneity is low ($\alpha = 0.1$), applying SCAFF-PD does not yield very large improvements compared to the existing algorithms. In the case of high data heterogeneity ($\alpha = 0.01$), our proposed algorithm largely improves the worst-20% accuracy performance on both datasets.

**Effect of $\rho$ in DRO.** To gain a better understanding of the empirical performance of our algorithm, we investigate the role of $\rho$ in DRO when applying our algorithm. We consider $\rho \in \{0.1, 0.2, 0.5\}$ and measure both the average and worst-20% accuracy during training. We present the results in Fig 3.

---

[1]First sort the clients by test accuracy, then select the lower $20\%$ of clients and compute the mean from this subset.

Table 1: The average and worst-20% top-1 accuracy of our algorithm (SCAFF-PD) vs. state-of-the-art federated learning algorithms evaluated on CIFAR100 and Tiny-ImageNet. The highest top-1 accuracy in each setting is highlighted in **bold**.

| Datasets | Methods | Non-i.i.d. degree | | | | | |
|---|---|---|---|---|---|---|---|
| | | $\alpha = 0.01$ | | $\alpha = 0.05$ | | $\alpha = 0.1$ | |
| | | average | worst-20% | average | worst-20% | average | worst-20% |
| CIFAR-100 | SCAFFOLD | 37.38 | 14.65 | 35.28 | 24.77 | 35.63 | 25.61 |
| | q-FFL (Li et al., 2019) | 32.27 | 13.88 | 36.92 | 24.66 | 38.83 | 30.36 |
| | AFL | 47.38 | 18.04 | **44.73** | 22.06 | **44.89** | 27.27 |
| | DRFA | 46.47 | 26.77 | 41.61 | 27.66 | 43.20 | 32.04 |
| | AgnosticFedAvg (Ro et al., 2021) | 46.02 | 21.52 | 37.06 | 27.97 | 38.81 | 29.31 |
| | *SCAFF-PD* | **49.03** | **29.30** | 42.06 | **28.37** | 43.69 | **32.77** |
| | | average | worst-20% | average | worst-20% | average | worst-20% |
| TinyImageNet | FedAvg | 33.66 | 18.18 | 31.53 | 23.46 | 35.08 | 27.61 |
| | SCAFFOLD | 31.79 | 15.85 | 30.43 | 22.57 | 34.58 | 27.33 |
| | q-FFL (Li et al., 2019) | 30.47 | 12.38 | 32.82 | 23.16 | 37.20 | 27.61 |
| | AFL | **45.32** | 18.65 | **45.54** | 28.02 | **46.11** | 29.50 |
| | DRFA | 36.80 | 22.32 | 37.39 | 28.38 | 37.39 | 28.38 |
| | AgnosticFedAvg (Ro et al., 2021) | 35.42 | 21.80 | 37.86 | 28.52 | 38.03 | 28.60 |
| | *SCAFF-PD* | 41.26 | **25.32** | 39.32 | **30.27** | 41.23 | **29.78** |

(a) Average accuracy.

(b) Worst-20% accuracy.

Figure 3: We study the effect of regularization term $\rho$ in our proposed algorithm SCAFF-PD. We measure both the average test accuracy (**a**) and worst-20% accuracy (**b**) during training. In addition, we include SCAFFOLD (orange dashed lines) as a baseline method for comparison.

We find that when $\rho$ is small, SCAFF-PD can achieve better fairness/robustness—the worst-20% accuracy significantly improves when we decrease the $\rho$ in SCAFF-PD. Meanwhile, the experimental results suggest that smaller $\rho$ leads to faster convergence w.r.t. worst-20% accuracy for our algorithm. On the other hand, when applying smaller $\rho$, the condition number of the min-max optimization problem becomes worse. Fortunately, our algorithm is guaranteed to achieve accelerated rates, making it particularly beneficial in scenarios where $\mu_{\boldsymbol{\lambda}}$ is small. As we have demonstrated in Fig 2, our proposed algorithm still converges relatively fast when $\rho$ is small.

In addition, we study the trade-off between average accuracy vs. worst-20% accuracy vs. best-20% accuracy for different algorithms. The results are summarized in Fig 4 (in Appendix C). Without sacrificing much on average accuracy and best-20% accuracy, our algorithm largely improves the worst-20% accuracy.

## 7  CONCLUSIONS

We have demonstrated the ability of SCAFF-PD to address challenges of fairness and robustness in federated learning. Theoretically, we obtained accelerated convergence rates for solving a wide class of federated DRO problems. Experimentally, we demonstrated strong empirical performance of SCAFF-PD on real-world datasets, improving upon existing approaches in both communication efficiency and model performance. An interesting future direction is the integration of DRO and privacy-preserving techniques in the context of federated learning, making SCAFF-PD applicable for a wider range of real-world applications. Another exciting direction is to explicitly integrate SCAFF-PD with game-theoretic mechanisms. Finally, studying the interplay between distributional robustness and personalization is an important open problem.

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
