## A   TECHNICAL LEMMAS

This section is dedicated to presenting several lemmas that serve as building blocks in proving the convergence of our proposed algorithms.

**Lemma A.1 (Perturbed strong convexity, Karimireddy et al. (2020))** *Suppose the function $f(\cdot)$ : $\mathcal{X} \to \mathbb{R}$ is $L$-smooth and $\mu$-strongly convex, then for any $x, y, z \in \mathcal{X}$,*

$$\langle \nabla f(x), z - y \rangle \geq f(z) - f(y) + \frac{\mu}{4}\|y - z\|^2 - L\|x - z\|^2. \tag{A.1}$$

We now present the lemma for analyzing the drift term.

**Lemma A.2 (Bounded drift)** *Suppose $\tau_r = J\eta_\ell \eta_g$, and $\eta_g \geq 1$, then we have*

$$\mathcal{E}_r \leq \frac{12\tau^2}{\eta_g^2}\mathbb{E}\left[\left\|\nabla_x \Phi(x^r, \lambda^{r+1})\right\|^2\right] + \frac{12\tau^2}{\eta_g^2}(1 + \chi)\zeta^2 + \frac{3\tau^2}{\eta_g^2 J}\zeta^2, \tag{A.2}$$

*where $\mathcal{E}_r$ is defined as*

$$\mathcal{E}_r = \frac{1}{J}\sum_{i=1}^{N}\sum_{j=1}^{J}\lambda_i \mathbb{E}\left[\|u_{i,j} - x\|^2\right], \tag{A.3}$$

*and $\chi$ is defined as*

$$\chi = \max_{\lambda \in \Lambda}\sum_{i=1}^{N}\lambda_i^2. \tag{A.4}$$

*Proof.* We omit the $r$ superscript in the following proof. Recall that the definition of $u_{i,j}$ in Algorithm 1, i.e.,

$$u_{i,j} = u_{i,j-1} - \eta_\ell\left(g_i(u_{i,j-1}) - \hat{c}_i + \hat{c}\right),$$

and we have

$$\begin{aligned}
\mathbb{E}\left[g_i(u_{i,j-1})\right] &= \nabla f_i(u_{i,j-1}), \\
\mathbb{E}\left[\hat{c}_i\right] &= \nabla f_i(x) = c_i, \\
\mathbb{E}\left[\hat{c}\right] &= \sum_{i=1}^{N}\lambda_i \nabla f_i(x) = c.
\end{aligned} \tag{A.5}$$

Then we can upper bound $\mathbb{E}\left[\|u_{i,j} - x\|^2\right]$ as follows,

$$\begin{aligned}
&\mathbb{E}\left[\|u_{i,j} - x\|^2\right] \\
&= \mathbb{E}\left[\|u_{i,j-1} - x - \eta_\ell(g_i(u_{i,j-1}) - \hat{c}_i + \hat{c})\|^2\right] \\
&= \mathbb{E}\left[\|u_{i,j-1} - x - \eta_\ell(\nabla f_i(u_{i,j-1}) - \hat{c}_i + \hat{c})\|^2\right] + \eta_\ell^2\mathbb{E}\left[\|g_i(u_{i,j-1}) - \nabla f_i(u_{i,j-1})\|^2\right] \\
&\leq \mathbb{E}\left[\|u_{i,j-1} - x - \eta_\ell(\nabla f_i(u_{i,j-1}) - \hat{c}_i + \hat{c})\|^2\right] + \eta_\ell^2\zeta^2 \\
&\leq \left(1 + \frac{1}{J-1}\right)\mathbb{E}\left[\|u_{i,j-1} - x\|^2\right] + J\eta_\ell^2\mathbb{E}\left[\|\nabla f_i(u_{i,j-1}) - \hat{c}_i + \hat{c}\|^2\right] + \eta_\ell^2\zeta^2 \\
&= \left(1 + \frac{1}{J-1}\right)\mathbb{E}\left[\|u_{i,j-1} - x\|^2\right] + \frac{\tau^2}{\eta_g^2 J}\mathbb{E}\left[\|\nabla f_i(u_{i,j-1}) - \hat{c}_i + \hat{c}\|^2\right] + \eta_\ell^2\zeta^2 \\
&\leq \left(1 + \frac{1}{J-1}\right)\mathbb{E}\left[\|u_{i,j-1} - x\|^2\right] + \frac{2\tau^2}{\eta_g^2 J}\mathbb{E}\left[\|\nabla f_i(u_{i,j-1}) - c_i + c\|^2\right] \\
&\quad + \frac{2\tau^2}{\eta_g^2 J}\mathbb{E}\left[\|\hat{c}_i - c_i + c - \hat{c}\|^2\right] + \eta_\ell^2\zeta^2 \\
&\leq \left(1 + \frac{1}{J-1}\right)\mathbb{E}\left[\|u_{i,j-1} - x\|^2\right] + \frac{2\tau^2}{\eta_g^2 J}\mathbb{E}\left[\|\nabla f_i(u_{i,j-1}) - c_i + c\|^2\right] \\
&\quad + \underbrace{\frac{4\tau^2}{\eta_g^2 J}\mathbb{E}\left[\|\hat{c}_i - c_i\|^2\right] + \frac{4\tau^2}{\eta_g^2 J}\mathbb{E}\left[\|\hat{c} - c\|^2\right] + \eta_\ell^2\zeta^2}_{\Gamma},
\end{aligned} \tag{A.6}$$

where $\Gamma = 0$ if the local gradients are deterministic. Next, we could first upper bound the term $\mathbb{E}\left[\|\boldsymbol{u}_{i,j} - \boldsymbol{x}\|^2\right]$ as

$$
\begin{aligned}
&\mathbb{E}\left[\|\boldsymbol{u}_{i,j} - \boldsymbol{x}\|^2\right] \\
&\leq \left(1 + \frac{1}{J-1}\right)\mathbb{E}\left[\|\boldsymbol{u}_{i,j-1} - \boldsymbol{x}\|^2\right] + \frac{4\tau^2}{\eta_g^2 J}\mathbb{E}\left[\|\nabla f_i(\boldsymbol{u}_{i,j-1}) - \boldsymbol{c}_i\|^2\right] + \frac{4\tau^2}{\eta_g^2 J}\mathbb{E}\left[\|\boldsymbol{c}\|^2\right] + \Gamma \\
&\leq \left(1 + \frac{1}{J-1} + \frac{4\tau^2 L_{\boldsymbol{xx}}^2}{\eta_g^2 J}\right)\mathbb{E}\left[\|\boldsymbol{u}_{i,j-1} - \boldsymbol{x}\|^2\right] + \frac{4\tau^2}{\eta_g^2 J}\mathbb{E}\left[\|\boldsymbol{c}\|^2\right] + \Gamma \\
&\leq \left(1 + \frac{2}{J-1}\right)\mathbb{E}\left[\|\boldsymbol{u}_{i,j-1} - \boldsymbol{x}\|^2\right] + \frac{4\tau^2}{\eta_g^2 J}\mathbb{E}\left[\|\boldsymbol{c}\|^2\right] + \Gamma,
\end{aligned}
\tag{A.7}
$$

where we apply the condition that $\frac{4\tau^2 L_{\boldsymbol{xx}}^2}{\eta_g^2 J} \leq \frac{1}{J-1}$. Then we have

$$
\begin{aligned}
\mathbb{E}\left[\|\boldsymbol{u}_{i,j} - \boldsymbol{x}\|^2\right] &\leq \sum_{i=1}^{j-1}\left(1 + \frac{2}{J-1}\right)^i\left(\frac{4\tau^2}{\eta_g^2 J}\mathbb{E}\left[\|\boldsymbol{c}\|^2\right] + \Gamma\right) \\
&\leq 3J\left(\frac{4\tau^2}{\eta_g^2 J}\mathbb{E}\left[\|\boldsymbol{c}\|^2\right] + \Gamma\right) = \frac{12\tau^2}{\eta_g^2}\mathbb{E}\left[\|\boldsymbol{c}\|^2\right] + 3J\Gamma.
\end{aligned}
\tag{A.8}
$$

Then the drift error $\mathcal{E}_r$ can be upper bounded as follows,

$$
\begin{aligned}
\mathcal{E}_r &= \frac{1}{J}\sum_{i=1}^{N}\sum_{j=1}^{J}\lambda_i\mathbb{E}\left[\|\boldsymbol{u}_{i,j} - \boldsymbol{x}\|^2\right] \\
&\leq \frac{1}{J}\sum_{i=1}^{N}\sum_{j=1}^{J}\lambda_i\frac{12\tau^2}{\eta_g^2}\mathbb{E}\left[\|\boldsymbol{c}\|^2\right] + \frac{1}{J}\sum_{i=1}^{N}\sum_{j=1}^{J}3\lambda_i J\Gamma \\
&= \frac{12\tau^2}{\eta_g^2}\mathbb{E}\left[\left\|\sum_{i=1}^{N}\lambda_i\nabla f_i(\boldsymbol{x})\right\|^2\right] + 3J\cdot\left(\frac{4\tau^2}{\eta_g^2 J}\sum_{i=1}^{N}\lambda_i\mathbb{E}\left[\|\hat{\boldsymbol{c}}_i - \boldsymbol{c}_i\|^2\right] + \frac{4\tau^2}{\eta_g^2 J}\mathbb{E}\left[\|\hat{\boldsymbol{c}} - \boldsymbol{c}\|^2\right] + \eta_\ell^2\zeta^2\right) \\
&\leq \frac{12\tau^2}{\eta_g^2}\mathbb{E}\left[\left\|\sum_{i=1}^{N}\lambda_i\nabla f_i(\boldsymbol{x})\right\|^2\right] + \left(\frac{12\tau^2}{\eta_g^2}\sum_{i=1}^{N}\lambda_i\mathbb{E}\left[\|\hat{\boldsymbol{c}}_i - \boldsymbol{c}_i\|^2\right] + \frac{12\tau^2}{\eta_g^2}\mathbb{E}\left[\|\hat{\boldsymbol{c}} - \boldsymbol{c}\|^2\right] + \frac{3\tau^2}{\eta_g^2 J}\zeta^2\right) \\
&\leq \frac{12\tau^2}{\eta_g^2}\mathbb{E}\left[\|\nabla_{\boldsymbol{x}}\Phi(\boldsymbol{x},\boldsymbol{\lambda})\|^2\right] + \frac{12\tau^2}{\eta_g^2}\zeta^2 + \frac{12\tau^2}{\eta_g^2}\underbrace{\left(\sum_{i=1}^{N}\lambda_i^2\right)}_{\leq\chi}\zeta^2 + \frac{3\tau^2}{\eta_g^2 J}\zeta^2 \\
&\leq \frac{12\tau^2}{\eta_g^2}\mathbb{E}\left[\|\nabla_{\boldsymbol{x}}\Phi(\boldsymbol{x},\boldsymbol{\lambda})\|^2\right] + \frac{12\tau^2}{\eta_g^2}(1+\chi)\zeta^2 + \frac{3\tau^2}{\eta_g^2 J}\zeta^2,
\end{aligned}
$$

which completes the proof. $\qquad\square$

The next lemma is useful in effectively controlling the drift term in our later analysis.

**Lemma A.3** *Suppose $\tau_r = J\eta_\ell\eta_g$, and $\eta_g \geq 1$, then we have*

$$
\frac{1}{\tau_r}\mathbb{E}\left[\|\boldsymbol{x}^{r+1} - \boldsymbol{x}^r\|^2\right] \geq -\tau_r L_{\boldsymbol{xx}}^2\mathcal{E}_r + \frac{\tau_r}{2}\left\|\nabla_{\boldsymbol{x}}\Phi(\boldsymbol{x}^r,\boldsymbol{\lambda}^{r+1})\right\|^2,
\tag{A.9}
$$

*where $\nabla_{\boldsymbol{x}}\Phi(\boldsymbol{x}^r,\boldsymbol{\lambda}^{r+1})$ is defined as*

$$
\nabla_{\boldsymbol{x}}\Phi(\boldsymbol{x}^r,\boldsymbol{\lambda}^{r+1}) = \sum_{i=1}^{N}\lambda_i^{r+1}\nabla f_i(\boldsymbol{x}^r).
\tag{A.10}
$$

*Proof.* In start with, we analyze $\frac{1}{4\tau_r}\|\boldsymbol{x}^{r+1} - \boldsymbol{x}^r\|^2$ based on the local updates, i.e.,

$$\frac{1}{\tau_r}\mathbb{E}\left[\|\boldsymbol{x}^{r+1} - \boldsymbol{x}^r\|^2\right]$$

$$= \tau_r\mathbb{E}\left[\left\|\frac{1}{J}\sum_{i=1}^{N}\sum_{j=1}^{J}\lambda_i^{r+1}g_i(\boldsymbol{u}_{i,j-1}^r)\right\|^2\right]$$

$$\geq \tau_r\mathbb{E}\left[\left\|\frac{1}{J}\sum_{i=1}^{N}\sum_{j=1}^{J}\lambda_i^{r+1}\nabla f_i(\boldsymbol{u}_{i,j-1}^r)\right\|^2\right]$$

$$\geq -\tau_r\mathbb{E}\left[\left\|\frac{1}{J}\sum_{i=1}^{N}\sum_{j=1}^{J}\lambda_i^{r+1}\nabla f_i(\boldsymbol{u}_{i,j-1}^r) - \sum_{i=1}^{N}\lambda_i^{r+1}\nabla f_i(\boldsymbol{x}^r)\right\|^2\right] + \frac{\tau_r}{2}\mathbb{E}\left[\|\nabla_{\boldsymbol{x}}\Phi(\boldsymbol{x}^r, \boldsymbol{\lambda}^{r+1})\|^2\right]$$

$$\geq -\tau_r\frac{1}{J}\sum_{i=1}^{N}\sum_{j=1}^{J}\lambda_i^{r+1}\mathbb{E}\left[\|\nabla f_i(\boldsymbol{u}_{i,j-1}^r) - \nabla f_i(\boldsymbol{x}^r)\|^2\right] + \frac{\tau_r}{2}\mathbb{E}\left[\|\nabla_{\boldsymbol{x}}\Phi(\boldsymbol{x}^r, \boldsymbol{\lambda}^{r+1})\|^2\right]$$

$$\geq -\tau_r L_{\boldsymbol{xx}}^2\underbrace{\frac{1}{J}\sum_{i=1}^{N}\sum_{j=1}^{J}\lambda_i^{r+1}\mathbb{E}\left[\|\boldsymbol{u}_{i,j-1}^r - \boldsymbol{x}^r\|^2\right]}_{\mathcal{E}_r} + \frac{\tau_r}{2}\mathbb{E}\left[\|\nabla_{\boldsymbol{x}}\Phi(\boldsymbol{x}^r, \boldsymbol{\lambda}^{r+1})\|^2\right]$$

$$= -\tau_r L_{\boldsymbol{xx}}^2\mathcal{E}_r + \frac{\tau_r}{2}\mathbb{E}\left[\|\nabla_{\boldsymbol{x}}\Phi(\boldsymbol{x}^r, \boldsymbol{\lambda}^{r+1})\|^2\right],$$

which completes the proof. $\qquad\square$

The next two lemmas focus on the primal update and dual update, respectively.

**Lemma A.4** *Suppose $\tau_r = J\eta_\ell\eta_g$, and $\eta_g \geq 1$, then we have*

$$\psi(\boldsymbol{\lambda}^{r+1}) - \langle \boldsymbol{s}^r, \boldsymbol{\lambda}^{r+1}\rangle$$
$$\leq \psi(\boldsymbol{\lambda}) - \langle \boldsymbol{s}^r, \boldsymbol{\lambda}\rangle + \frac{1}{\sigma_r}\left[D(\boldsymbol{\lambda}, \boldsymbol{\lambda}^r) - D(\boldsymbol{\lambda}, \boldsymbol{\lambda}^{r+1}) - D(\boldsymbol{\lambda}^{r+1}, \boldsymbol{\lambda}^r)\right] - \frac{\mu_{\boldsymbol{\lambda}}}{2}\|\boldsymbol{\lambda}_{r+1} - \boldsymbol{\lambda}\|^2. \tag{A.11}$$

*Proof.* Based on Property 1 in Tseng (2008), and $D(\boldsymbol{\lambda}, \boldsymbol{\lambda}') = \|\boldsymbol{\lambda} - \boldsymbol{\lambda}'\|^2/2$. $\qquad\square$

**Lemma A.5** *Suppose $\tau_r = J\eta_\ell\eta_g$, and $\eta_g \geq 1$, then we have*

$$\mathbb{E}\left[\langle\Delta\boldsymbol{x}^r, \boldsymbol{x}^{r+1} - \boldsymbol{x}\rangle\right] \leq \frac{1}{\tau_r}\mathbb{E}\left[D(\boldsymbol{x}, \boldsymbol{x}^r) - D(\boldsymbol{x}, \boldsymbol{x}^{r+1}) - D(\boldsymbol{x}^{r+1}, \boldsymbol{x}^r)\right], \tag{A.12}$$

*and*

$$\mathbb{E}\left[\langle\Delta\boldsymbol{x}^r, \boldsymbol{x}^{r+1} - \boldsymbol{x}\rangle\right]$$
$$= \underbrace{\mathbb{E}\left[\langle\Delta\boldsymbol{x}^r, \boldsymbol{x}^r - \boldsymbol{x}\rangle\right]}_{\mathcal{T}_1} + \underbrace{\mathbb{E}\left[\langle\widetilde{\Delta}\boldsymbol{x}^r, \boldsymbol{x}^{r+1} - \boldsymbol{x}^r\rangle\right]}_{\mathcal{T}_2} + \underbrace{\mathbb{E}\left[\langle\Delta\boldsymbol{x}^r - \widetilde{\Delta}\boldsymbol{x}^r, \boldsymbol{x}^{r+1} - \boldsymbol{x}^r\rangle\right]}_{\mathcal{T}_3}, \tag{A.13}$$

*where*

$$\mathcal{T}_1 \geq \mathbb{E}\left[\Phi(\boldsymbol{x}^r, \boldsymbol{\lambda}^{r+1}) - \Phi(\boldsymbol{x}, \boldsymbol{\lambda}^{r+1})\right] + \frac{\mu_{\boldsymbol{x}}}{4}\mathbb{E}\left[\|\boldsymbol{x}^r - \boldsymbol{x}\|^2\right] - L_{\boldsymbol{xx}}\mathcal{E}_r,$$
$$\mathcal{T}_2 \geq \mathbb{E}\left[\Phi(\boldsymbol{x}^{r+1}, \boldsymbol{\lambda}^{r+1}) - \Phi(\boldsymbol{x}^r, \boldsymbol{\lambda}^{r+1})\right] - 2L_{\boldsymbol{xx}}\mathbb{E}\left[\|\boldsymbol{x}^{r+1} - \boldsymbol{x}^r\|^2\right] - 2L_{\boldsymbol{xx}}\mathcal{E}_r, \tag{A.14}$$
$$\mathcal{T}_3 \geq -\frac{2\chi\tau_r}{J}\zeta^2 - \frac{1}{4\tau_r}\mathbb{E}\left[D(\boldsymbol{x}^{r+1}, \boldsymbol{x}^r)\right],$$

*and $\Delta\boldsymbol{x}^r$ and $\widetilde{\Delta}\boldsymbol{x}^r$ are defined as*

$$\Delta\boldsymbol{x}^r = \frac{1}{J}\sum_{i=1}^{N}\sum_{j=1}^{J}\lambda_i^{r+1}g_i(\boldsymbol{u}_{i,j-1}^r), \quad \widetilde{\Delta}\boldsymbol{x}^r = \mathbb{E}[\Delta\boldsymbol{x}^r] = \frac{1}{J}\sum_{i=1}^{N}\sum_{j=1}^{J}\lambda_i^{r+1}\mathbb{E}\left[\nabla f_i(\boldsymbol{u}_{i,j-1}^r)\right]. \tag{A.15}$$

*Proof.* Based on Property 1 in Tseng (2008), for the update step of $\boldsymbol{x}^{r+1}$, we have

$$\mathbb{E}\left[\left\langle\sum_{i=1}^{N}\lambda_i^{r+1}\Delta\boldsymbol{u}_i^r, \boldsymbol{x}^{r+1}-\boldsymbol{x}\right\rangle\right] \leq \frac{1}{\tau_r}\mathbb{E}\left[D(\boldsymbol{x},\boldsymbol{x}^r)-D(\boldsymbol{x},\boldsymbol{x}^{r+1})-D(\boldsymbol{x}^{r+1},\boldsymbol{x}^r)\right], \quad \text{(A.16)}$$

then we analyze the term $\Delta\boldsymbol{x}^r = \sum_{i=1}^{N}\lambda_i^{r+1}\Delta\boldsymbol{u}_i^r$, i.e.,

$$\begin{aligned}
\sum_{i=1}^{N}\lambda_i^{r+1}\Delta\boldsymbol{u}_i^r &= \frac{1}{J}\sum_{i=1}^{N}\sum_{j=1}^{J}\lambda_i^{r+1}\left(g_i(\boldsymbol{u}_{i,j-1}^r)-\hat{\boldsymbol{c}}_i^r+\hat{\boldsymbol{c}}^r\right) \\
&= \frac{1}{J}\sum_{i=1}^{N}\sum_{j=1}^{J}\lambda_i^{r+1}g_i(\boldsymbol{u}_{i,j-1}^r)-\frac{1}{J}\sum_{i=1}^{N}\sum_{j=1}^{J}\lambda_i^{r+1}\hat{\boldsymbol{c}}_i^r+\frac{1}{J}\sum_{i=1}^{N}\sum_{j=1}^{J}\lambda_i^{r+1}\hat{\boldsymbol{c}}^r \quad \text{(A.17)} \\
&= \frac{1}{J}\sum_{i=1}^{N}\sum_{j=1}^{J}\lambda_i^{r+1}g_i(\boldsymbol{u}_{i,j-1}^r).
\end{aligned}$$

Next we decompose $\mathbb{E}\left[\left\langle\Delta\boldsymbol{x}^r, \boldsymbol{x}^{r+1}-\boldsymbol{x}\right\rangle\right]$ as follows,

$$\begin{aligned}
&\mathbb{E}\left[\left\langle\Delta\boldsymbol{x}^r, \boldsymbol{x}^{r+1}-\boldsymbol{x}\right\rangle\right] \\
&= \underbrace{\mathbb{E}\left[\left\langle\Delta\boldsymbol{x}^r, \boldsymbol{x}^r-\boldsymbol{x}\right\rangle\right]}_{\mathcal{T}_1}+\underbrace{\mathbb{E}\left[\left\langle\widetilde{\Delta}\boldsymbol{x}^r, \boldsymbol{x}^{r+1}-\boldsymbol{x}^r\right\rangle\right]}_{\mathcal{T}_2}+\underbrace{\mathbb{E}\left[\left\langle\Delta\boldsymbol{x}^r-\widetilde{\Delta}\boldsymbol{x}^r, \boldsymbol{x}^{r+1}-\boldsymbol{x}^r\right\rangle\right]}_{\mathcal{T}_3}. \quad \text{(A.18)}
\end{aligned}$$

We then analyze the upper bound for $|\mathcal{T}_3|$, i.e.,

$$\begin{aligned}
|\mathcal{T}_3| &= \mathbb{E}\left[\left|\langle\Delta\boldsymbol{x}^r-\widetilde{\Delta}\boldsymbol{x}^r, \boldsymbol{x}^{r+1}-\boldsymbol{x}^r\rangle\right|\right] \\
&\leq \tau_r\mathbb{E}\left[\|\Delta\boldsymbol{x}^r-\widetilde{\Delta}\boldsymbol{x}^r\|^2\right]+\frac{1}{4\tau_r}\mathbb{E}\left[\|\boldsymbol{x}^{r+1}-\boldsymbol{x}^r\|^2\right] \quad \text{(A.19)} \\
&\leq \frac{2\chi\tau_r}{J}\zeta^2+\frac{1}{4\tau_r}\mathbb{E}\left[\|\boldsymbol{x}^{r+1}-\boldsymbol{x}^r\|^2\right].
\end{aligned}$$

Next, we analyze term $\mathcal{T}_1$, i.e.,

$$\begin{aligned}
\mathcal{T}_1 &= \mathbb{E}\left[\left\langle\Delta\boldsymbol{x}^r, \boldsymbol{x}^r-\boldsymbol{x}\right\rangle\right] = \mathbb{E}\left[\left\langle\widetilde{\Delta}\boldsymbol{x}^r, \boldsymbol{x}^r-\boldsymbol{x}\right\rangle\right] \\
&= \mathbb{E}\left[\left\langle\frac{1}{J}\sum_{i=1}^{N}\sum_{j=1}^{J}\lambda_i^{r+1}\nabla f_i(\boldsymbol{u}_{i,j-1}^r), \boldsymbol{x}^r-\boldsymbol{x}\right\rangle\right] \\
&\geq \frac{1}{J}\sum_{i,j}\lambda_i^{r+1}\mathbb{E}\left[f_i(\boldsymbol{x}^r)-f_i(\boldsymbol{x})+\frac{\mu_{\boldsymbol{x}}}{4}\|\boldsymbol{x}^r-\boldsymbol{x}\|^2-L_{\boldsymbol{x}\boldsymbol{x}}\|\boldsymbol{u}_{i,j-1}^r-\boldsymbol{x}^r\|^2\right] \\
&= \mathbb{E}\left[\sum_i\lambda_i^{r+1}f_i(\boldsymbol{x}^r)-\sum_i\lambda_i^{r+1}f_i(\boldsymbol{x})+\frac{\mu_{\boldsymbol{x}}}{4}\|\boldsymbol{x}^r-\boldsymbol{x}\|^2-L_{\boldsymbol{x}\boldsymbol{x}}\frac{1}{J}\sum_{i,j}\lambda_i^{r+1}\|\boldsymbol{u}_{i,j-1}^r-\boldsymbol{x}^r\|^2\right] \\
&= \mathbb{E}\left[\Phi(\boldsymbol{x}^r,\boldsymbol{\lambda}^{r+1})-\Phi(\boldsymbol{x},\boldsymbol{\lambda}^{r+1})\right]+\frac{\mu_{\boldsymbol{x}}}{4}\mathbb{E}\left[\|\boldsymbol{x}^r-\boldsymbol{x}\|^2\right]-L_{\boldsymbol{x}\boldsymbol{x}}\mathcal{E}_r,
\end{aligned}$$

where we apply the perturbed strong convexity lemma (Lemma A.1) for the first inequality.

We then analyze term $\mathcal{T}_2$,

$$
\begin{aligned}
\mathcal{T}_2 &= \mathbb{E}\left[\left\langle \widetilde{\Delta}\boldsymbol{x}^r, \boldsymbol{x}^{r+1} - \boldsymbol{x}^r \right\rangle\right] \\
&= \mathbb{E}\left[\left\langle \frac{1}{J}\sum_{i=1}^{N}\sum_{j=1}^{J}\lambda_i^{r+1}\nabla f_i(\boldsymbol{u}_{i,j-1}^r), \boldsymbol{x}^{r+1} - \boldsymbol{x}^r \right\rangle\right] \\
&\geq \mathbb{E}\left[\Phi(\boldsymbol{x}^{r+1}, \boldsymbol{\lambda}^{r+1}) - \Phi(\boldsymbol{x}^r, \boldsymbol{\lambda}^{r+1}) + \frac{\mu_{\boldsymbol{x}}}{4}\|\boldsymbol{x}^{r+1} - \boldsymbol{x}^r\|^2 - \frac{L_{\boldsymbol{xx}}}{J}\sum_{i,j}\lambda_i^{r+1}\|\boldsymbol{u}_{i,j-1}^r - \boldsymbol{x}^{r+1}\|^2\right] \\
&\geq \mathbb{E}\left[\Phi(\boldsymbol{x}^{r+1}, \boldsymbol{\lambda}^{r+1}) - \Phi(\boldsymbol{x}^r, \boldsymbol{\lambda}^{r+1}) + \left(\frac{\mu_{\boldsymbol{x}}}{4} - 2L_{\boldsymbol{xx}}\right)\|\boldsymbol{x}^{r+1} - \boldsymbol{x}^r\|^2\right] - 2L_{\boldsymbol{xx}}\mathcal{E}_r \\
&\geq \mathbb{E}\left[\Phi(\boldsymbol{x}^{r+1}, \boldsymbol{\lambda}^{r+1}) - \Phi(\boldsymbol{x}^r, \boldsymbol{\lambda}^{r+1})\right] - 2L_{\boldsymbol{xx}}\mathbb{E}\left[\|\boldsymbol{x}^{r+1} - \boldsymbol{x}^r\|^2\right] - 2L_{\boldsymbol{xx}}\mathcal{E}_r,
\end{aligned}
$$

where we apply the perturbed strong convexity lemma (Lemma A.1) for the first inequality and apply $\|\boldsymbol{x} + \boldsymbol{y}\|^2 \leq 2\|\boldsymbol{x}\|^2 + 2\|\boldsymbol{y}\|^2$ for the second inequality. This completes our proof. $\square$

## B CONVERGENCE OF SCAFF-PD

In this section, we present the missing proofs in Section 5. Specifically, Section B.1 contains the proofs for the strongly convex-concave setting (Section 5.1), while Section B.2 includes the proofs for the strongly convex-strongly concave setting (Section 5.2).

### B.1 PROOFS – STRONGLY-CONVEX-CONCAVE (SC-C) SETTING

In this subsection, we first present the technical lemma in Section B.1.1. Next, we analyze how to set the step size related parameters in Section B.1.2. Finally, we prove Theorem 5.1 in Section B.1.3.

#### B.1.1 TECHNICAL LEMMA

**Lemma B.1** *If we set the step size in Algorithm 1 as $\tau_r \cdot L_{\boldsymbol{xx}} \leq 1$, and the parameters of Algorithm 1 satisfy Condition 5.1, then for any $\boldsymbol{x}, \boldsymbol{\lambda}$ we have*

$$\mathbb{E}\left[F(\boldsymbol{x}^{r+1}, \boldsymbol{\lambda}) - F(\boldsymbol{x}, \boldsymbol{\lambda}^{r+1})\right] \leq -Z_{r+1} + V_r + \Delta_r + C\tau_r\zeta^2, \tag{B.1}$$

*where $Z_{r+1}, V_r, \Delta_r$ are defined as*

$$Z_{r+1} = \mathbb{E}\left[\langle \boldsymbol{q}^{r+1}, \boldsymbol{\lambda}^{r+1} - \boldsymbol{\lambda}\rangle + \frac{1}{2\sigma_r}\|\boldsymbol{\lambda}^{r+1} - \boldsymbol{\lambda}\|^2 + \left(\frac{1}{2\tau_r} + \frac{\mu_{\boldsymbol{x}}}{8}\right)\|\boldsymbol{x}^{r+1} - \boldsymbol{x}\|^2 + \frac{1}{2\alpha_{r+1}}\|\boldsymbol{q}^{r+1}\|^2\right],$$

$$V_r = \mathbb{E}\left[\theta_r\langle \boldsymbol{q}^r, \boldsymbol{\lambda}^r - \boldsymbol{\lambda}\rangle + \frac{1}{2\sigma_r}\|\boldsymbol{\lambda}^r - \boldsymbol{\lambda}\|^2 + \frac{1}{2\tau_r}\|\boldsymbol{x}^r - \boldsymbol{x}\|^2 + \frac{\theta_r}{2\alpha_r}\|\boldsymbol{q}^r\|^2\right], \tag{B.2}$$

$$\Delta_r = \mathbb{E}\left[\left(\frac{\alpha_r\theta_r}{2} - \frac{1}{2\sigma_r}\right)\|\boldsymbol{\lambda}^{r+1} - \boldsymbol{\lambda}^r\|^2 + \left(\frac{L_{\boldsymbol{\lambda x}}^2}{2\alpha_{r+1}} + 3L_{\boldsymbol{xx}} - \frac{1}{4\tau_r}\right)\|\boldsymbol{x}^{r+1} - \boldsymbol{x}^r\|^2\right],$$

*$\boldsymbol{q}^r$ is defined as*

$$\boldsymbol{q}^r = \nabla_{\boldsymbol{\lambda}}\Phi(\boldsymbol{x}^r, \boldsymbol{\lambda}^r) - \nabla_{\boldsymbol{\lambda}}\Phi(\boldsymbol{x}^{r-1}, \boldsymbol{\lambda}^{r-1}), \tag{B.3}$$

*and $C \geq 0$ is a constant.*

*Proof.* To start with, by applying Lemma A.4, we have

$$\psi(\boldsymbol{\lambda}^{r+1}) - \langle \boldsymbol{s}^r, \boldsymbol{\lambda}^{r+1} - \boldsymbol{\lambda}\rangle \leq \psi(\boldsymbol{\lambda}) + \underbrace{\frac{1}{\sigma_r}\left[D(\boldsymbol{\lambda}, \boldsymbol{\lambda}^r) - D(\boldsymbol{\lambda}, \boldsymbol{\lambda}^{r+1}) - D(\boldsymbol{\lambda}^{r+1}, \boldsymbol{\lambda}^r)\right]}_{B_r}, \tag{B.4}$$

where we define $B_r$ as

$$B_r = \frac{1}{\sigma_r}\left[D(\boldsymbol{\lambda}, \boldsymbol{\lambda}^r) - D(\boldsymbol{\lambda}, \boldsymbol{\lambda}^{r+1}) - D(\boldsymbol{\lambda}^{r+1}, \boldsymbol{\lambda}^r)\right]. \tag{B.5}$$

Then by applying Lemma A.5, for the update step of $\boldsymbol{x}^{r+1}$, we have

$$\mathbb{E}\left[\langle \Delta\boldsymbol{x}^r, \boldsymbol{x}^{r+1} - \boldsymbol{x}\rangle\right] \leq \frac{1}{\tau_r}\mathbb{E}\left[D(\boldsymbol{x}, \boldsymbol{x}^r) - D(\boldsymbol{x}, \boldsymbol{x}^{r+1}) - D(\boldsymbol{x}^{r+1}, \boldsymbol{x}^r)\right], \tag{B.6}$$

where $\Delta\boldsymbol{x}^r$ is defined in Eq. (A.15). Then we decompose the term $\mathbb{E}[\langle \Delta\boldsymbol{x}^r, \boldsymbol{x}^{r+1} - \boldsymbol{x}\rangle]$ as follows,

$$\mathbb{E}\left[\langle \Delta\boldsymbol{x}^r, \boldsymbol{x}^{r+1} - \boldsymbol{x}\rangle\right]$$
$$= \underbrace{\mathbb{E}\left[\langle \Delta\boldsymbol{x}^r, \boldsymbol{x}^r - \boldsymbol{x}\rangle\right]}_{\mathcal{T}_1} + \underbrace{\mathbb{E}\left[\langle \widetilde{\Delta}\boldsymbol{x}^r, \boldsymbol{x}^{r+1} - \boldsymbol{x}^r\rangle\right]}_{\mathcal{T}_2} + \underbrace{\mathbb{E}\left[\langle \Delta\boldsymbol{x}^r - \widetilde{\Delta}\boldsymbol{x}^r, \boldsymbol{x}^{r+1} - \boldsymbol{x}^r\rangle\right]}_{\mathcal{T}_3} \tag{B.7}$$
$$= \mathcal{T}_1 + \mathcal{T}_2 + \mathcal{T}_3,$$

and by Lemma A.5,

$$\mathcal{T}_1 \geq \mathbb{E}\left[\Phi(\boldsymbol{x}^r, \boldsymbol{\lambda}^{r+1}) - \Phi(\boldsymbol{x}, \boldsymbol{\lambda}^{r+1})\right] + \frac{\mu_{\boldsymbol{x}}}{4}\mathbb{E}\left[\|\boldsymbol{x}^r - \boldsymbol{x}\|^2\right] - L_{\boldsymbol{xx}}\mathcal{E}_r,$$

$$\mathcal{T}_2 \geq \mathbb{E}\left[\Phi(\boldsymbol{x}^{r+1}, \boldsymbol{\lambda}^{r+1}) - \Phi(\boldsymbol{x}^r, \boldsymbol{\lambda}^{r+1})\right] - 2L_{\boldsymbol{xx}}\mathbb{E}\left[\|\boldsymbol{x}^{r+1} - \boldsymbol{x}^r\|^2\right] - 2L_{\boldsymbol{xx}}\mathcal{E}_r, \tag{B.8}$$

$$\mathcal{T}_3 \geq -\frac{2\chi\tau_r}{J}\zeta^2 - \frac{1}{4\tau_r}\mathbb{E}\left[D(\boldsymbol{x}^{r+1}, \boldsymbol{x}^r)\right].$$

Therefore, by combining Eq. (B.6) and Eq. (B.8), we have

$$
\begin{aligned}
&\mathbb{E}\left[\Phi(\boldsymbol{x}^{r+1}, \boldsymbol{\lambda}) - \Phi(\boldsymbol{x}, \boldsymbol{\lambda}^{r+1})\right] \\
&\leq \mathbb{E}\left[\Phi(\boldsymbol{x}^{r+1}, \boldsymbol{\lambda}) - \Phi(\boldsymbol{x}^{r+1}, \boldsymbol{\lambda}^{r+1}) + \Phi(\boldsymbol{x}^{r+1}, \boldsymbol{\lambda}^{r+1}) - \Phi(\boldsymbol{x}^r, \boldsymbol{\lambda}^{r+1})\right] - \mathcal{T}_2 - \mathcal{T}_3 \\
&\quad + \frac{1}{\tau_r}\mathbb{E}\left[\mathrm{D}(\boldsymbol{x}, \boldsymbol{x}^r) - \mathrm{D}(\boldsymbol{x}, \boldsymbol{x}^{r+1}) - \mathrm{D}(\boldsymbol{x}^{r+1}, \boldsymbol{x}^r)\right] - \frac{\mu_{\boldsymbol{x}}}{4}\mathbb{E}\left[\|\boldsymbol{x}^r - \boldsymbol{x}\|^2\right] + L_{\boldsymbol{xx}}\mathcal{E}_r \\
&\leq \mathbb{E}\left[\Phi(\boldsymbol{x}^{r+1}, \boldsymbol{\lambda}) - \Phi(\boldsymbol{x}^{r+1}, \boldsymbol{\lambda}^{r+1})\right] + \frac{1}{\tau_r}\mathbb{E}\left[\mathrm{D}(\boldsymbol{x}, \boldsymbol{x}^r) - \mathrm{D}(\boldsymbol{x}, \boldsymbol{x}^{r+1}) - \frac{1}{2}\mathrm{D}(\boldsymbol{x}^{r+1}, \boldsymbol{x}^r)\right] \\
&\quad - \frac{\mu_{\boldsymbol{x}}}{4}\mathbb{E}\left[\|\boldsymbol{x}^r - \boldsymbol{x}\|^2\right] + 2L_{\boldsymbol{xx}}\mathbb{E}\left[\|\boldsymbol{x}^{r+1} - \boldsymbol{x}^r\|^2\right] + 3L_{\boldsymbol{xx}}\mathcal{E}_r + \frac{2\chi\,\tau_r}{J}\zeta^2 \\
&\leq \mathbb{E}\left[\Phi(\boldsymbol{x}^{r+1}, \boldsymbol{\lambda}) - \Phi(\boldsymbol{x}^{r+1}, \boldsymbol{\lambda}^{r+1})\right] + 3L_{\boldsymbol{xx}}\mathbb{E}\left[\|\boldsymbol{x}^{r+1} - \boldsymbol{x}^r\|^2\right] + 3L_{\boldsymbol{xx}}\mathcal{E}_r + \frac{2\chi\,\tau_r}{J}\zeta^2 \\
&\quad + \underbrace{\frac{1}{\tau_r}\mathbb{E}\left[\mathrm{D}(\boldsymbol{x}, \boldsymbol{x}^r) - \mathrm{D}(\boldsymbol{x}, \boldsymbol{x}^{r+1}) - \frac{1}{2}\mathrm{D}(\boldsymbol{x}^{r+1}, \boldsymbol{x}^r)\right] - \frac{\mu_{\boldsymbol{x}}}{8}\mathbb{E}\left[\|\boldsymbol{x}^{r+1} - \boldsymbol{x}\|^2\right]}_{A_r} \\
&= \mathbb{E}\left[\Phi(\boldsymbol{x}^{r+1}, \boldsymbol{\lambda}) - \Phi(\boldsymbol{x}^{r+1}, \boldsymbol{\lambda}^{r+1})\right] + 3L_{\boldsymbol{xx}}\mathbb{E}\left[\|\boldsymbol{x}^{r+1} - \boldsymbol{x}^r\|^2\right] + 3L_{\boldsymbol{xx}}\mathcal{E}_r + \frac{2\chi\,\tau_r}{J}\zeta^2 + A_r,
\end{aligned}
\tag{B.9}
$$

where we apply the lower bound of $\mathcal{T}_2$ and $\mathcal{T}_3$ (Eq. (B.8)) for the second inequality, and apply $-\|\boldsymbol{x}\|^2 \leq \|\boldsymbol{y}\|^2 - \frac{1}{2}\|\boldsymbol{x} + \boldsymbol{y}\|^2$ for the third inequality, and apply $\mu_{\boldsymbol{x}} \leq L_{\boldsymbol{xx}}$ for the last inequality. We define $A_r$ as

$$
A_r = \frac{1}{\tau_r}\mathbb{E}\left[\mathrm{D}(\boldsymbol{x}, \boldsymbol{x}^r) - \mathrm{D}(\boldsymbol{x}, \boldsymbol{x}^{r+1}) - \frac{1}{2}\mathrm{D}(\boldsymbol{x}^{r+1}, \boldsymbol{x}^r)\right] - \frac{\mu_{\boldsymbol{x}}}{8}\mathbb{E}\left[\|\boldsymbol{x}^{r+1} - \boldsymbol{x}\|^2\right].
\tag{B.10}
$$

Next, we apply the concavity of $\Phi(\boldsymbol{x}^{r+1}, \cdot)$ and combining the above two steps, for the $\boldsymbol{x}$-update we have

$$
\begin{aligned}
&\mathbb{E}\left[\Phi(\boldsymbol{x}^{r+1}, \boldsymbol{\lambda}) - \Phi(\boldsymbol{x}, \boldsymbol{\lambda}^{r+1})\right] \\
&\leq \mathbb{E}[\langle\nabla_{\boldsymbol{\lambda}}\Phi(\boldsymbol{x}^{r+1}, \boldsymbol{\lambda}^{r+1}), \boldsymbol{\lambda} - \boldsymbol{\lambda}^{r+1}\rangle] + A_r + 3L_{\boldsymbol{xx}}\mathbb{E}\left[\|\boldsymbol{x}^{r+1} - \boldsymbol{x}^r\|^2\right] + 3L_{\boldsymbol{xx}}\mathcal{E}_r + \frac{2\chi\,\tau_r}{J}\zeta^2,
\end{aligned}
\tag{B.11}
$$

By combining the inequality of $\boldsymbol{\lambda}$-update (Eq. (B.4)) and $\boldsymbol{x}$-update (Eq. (B.11)), we can get

$$
\begin{aligned}
&\mathbb{E}\left[F(\boldsymbol{x}^{r+1}, \boldsymbol{\lambda}) - F(\boldsymbol{x}, \boldsymbol{\lambda}^{r+1})\right] \\
&= \mathbb{E}\left[\left(\Phi(\boldsymbol{x}^{r+1}, \boldsymbol{\lambda}) - \psi(\boldsymbol{\lambda})\right) - \left(\Phi(\boldsymbol{x}, \boldsymbol{\lambda}^{r+1}) - \psi(\boldsymbol{\lambda}^{r+1})\right)\right] \\
&\leq \mathbb{E}\left[\langle\nabla_{\boldsymbol{\lambda}}\Phi(\boldsymbol{x}^{r+1}, \boldsymbol{\lambda}^{r+1}), \boldsymbol{\lambda} - \boldsymbol{\lambda}^{r+1}\rangle\right] + \langle\boldsymbol{s}^r, \boldsymbol{\lambda}^{r+1} - \boldsymbol{\lambda}\rangle + A_r + B_r \\
&\quad + 3L_{\boldsymbol{xx}}\mathcal{E}_r + 3L_{\boldsymbol{xx}}\mathbb{E}\left[\|\boldsymbol{x}^{r+1} - \boldsymbol{x}^r\|^2\right] + \frac{2\chi\,\tau_r}{J}\zeta^2 \\
&= -\mathbb{E}\left[\langle\boldsymbol{q}^{r+1}, \boldsymbol{\lambda}^{r+1} - \boldsymbol{\lambda}\rangle\right] + \theta_r\mathbb{E}\left[\langle\boldsymbol{q}^r, \boldsymbol{\lambda}^{r+1} - \boldsymbol{\lambda}\rangle\right] + A_r + B_r \\
&\quad + 3L_{\boldsymbol{xx}}\mathcal{E}_r + 3L_{\boldsymbol{xx}}\mathbb{E}\left[\|\boldsymbol{x}^{r+1} - \boldsymbol{x}^r\|^2\right] + \frac{2\chi\,\tau_r}{J}\zeta^2 \\
&= -\langle\mathbb{E}\left[\boldsymbol{q}^{r+1}, \boldsymbol{\lambda}^{r+1} - \boldsymbol{\lambda}\right] + \theta_r\mathbb{E}[\langle\boldsymbol{q}^r, \boldsymbol{\lambda}^r - \boldsymbol{\lambda}\rangle] + \theta_r\mathbb{E}\left[\langle\boldsymbol{q}^r, \boldsymbol{\lambda}^{r+1} - \boldsymbol{\lambda}^r\rangle\right] \\
&\quad + A_r + B_r + 3L_{\boldsymbol{xx}}\mathcal{E}_r + 3L_{\boldsymbol{xx}}\mathbb{E}\left[\|\boldsymbol{x}^{r+1} - \boldsymbol{x}^r\|^2\right] + \frac{2\chi\,\tau_r}{J}\zeta^2,
\end{aligned}
\tag{B.12}
$$

where we apply Eq. (B.11) for the first inequality, and the definition of $\boldsymbol{q}^r$ for the second equality, and $\boldsymbol{q}^r$ is defined as

$$
\boldsymbol{q}^r = \nabla_{\boldsymbol{\lambda}}\Phi(\boldsymbol{x}^r, \boldsymbol{\lambda}^r) - \nabla_{\boldsymbol{\lambda}}\Phi(\boldsymbol{x}^{r-1}, \boldsymbol{\lambda}^{r-1}).
\tag{B.13}
$$

The term $\theta^r\langle\boldsymbol{q}^r, \boldsymbol{\lambda}^{r+1} - \boldsymbol{\lambda}^r\rangle$ can be upper bounded as

$$
\begin{aligned}
\theta_r\langle\boldsymbol{q}^r, \boldsymbol{\lambda}^{r+1} - \boldsymbol{\lambda}^r\rangle &= \theta_r\langle\nabla_{\boldsymbol{\lambda}}\Phi(\boldsymbol{x}^r, \boldsymbol{\lambda}^r) - \nabla_{\boldsymbol{\lambda}}\Phi(\boldsymbol{x}^{r-1}, \boldsymbol{\lambda}^{r-1}), \boldsymbol{\lambda}^{r+1} - \boldsymbol{\lambda}^r\rangle \\
&= \theta_r\langle\nabla_{\boldsymbol{\lambda}}\Phi(\boldsymbol{x}^r, \boldsymbol{\lambda}^r) - \nabla_{\boldsymbol{\lambda}}\Phi(\boldsymbol{x}^{r-1}, \boldsymbol{\lambda}^r), \boldsymbol{\lambda}^{r+1} - \boldsymbol{\lambda}^r\rangle \\
&\leq \frac{\theta_r}{2\alpha_r}\|\nabla_{\boldsymbol{\lambda}}\Phi(\boldsymbol{x}^r, \boldsymbol{\lambda}^r) - \nabla_{\boldsymbol{\lambda}}\Phi(\boldsymbol{x}^{r-1}, \boldsymbol{\lambda}^r)\|^2 + \frac{\alpha_r\theta_r}{2}\|\boldsymbol{\lambda}^{r+1} - \boldsymbol{\lambda}^r\|^2
\end{aligned}
\tag{B.14}
$$

where we apply $\nabla_{\boldsymbol{\lambda}}\Phi(\boldsymbol{x}^{r-1}, \boldsymbol{\lambda}^r) = \nabla_{\boldsymbol{\lambda}}\Phi(\boldsymbol{x}^{r-1}, \boldsymbol{\lambda}^{r-1})$ (because the $\Phi$ is linear in $\boldsymbol{\lambda}$) for the second equality, and apply the smoothness assumption

$$
\|\boldsymbol{q}^r\| = \|\nabla_{\boldsymbol{\lambda}}\Phi(\boldsymbol{x}^r, \boldsymbol{\lambda}^r) - \nabla_{\boldsymbol{\lambda}}\Phi(\boldsymbol{x}^{r-1}, \boldsymbol{\lambda}^r)\| \leq L_{\boldsymbol{\lambda x}}\|\boldsymbol{x}^r - \boldsymbol{x}^{r-1}\|
$$

in the second inequality. Then by combining Eq. (B.14) and Eq. (B.12), we have

$$\mathbb{E}\left[F(\boldsymbol{x}^{r+1}, \boldsymbol{\lambda}) - F(\boldsymbol{x}, \boldsymbol{\lambda}^{r+1})\right]$$

$$\leq -\underbrace{\mathbb{E}\left[\langle \boldsymbol{q}^{r+1}, \boldsymbol{\lambda}^{r+1} - \boldsymbol{\lambda}\rangle + \frac{1}{\sigma_r}\mathrm{D}(\boldsymbol{\lambda}, \boldsymbol{\lambda}^{r+1}) + \frac{1}{\tau_r}\mathrm{D}(\boldsymbol{x}, \boldsymbol{x}^{r+1}) + \frac{\mu_{\boldsymbol{x}}}{8}\|\boldsymbol{x}^{r+1} - \boldsymbol{x}\|^2 + \frac{1}{2\alpha_{r+1}}\|\boldsymbol{q}^{r+1}\|^2\right]}_{Z_{r+1}}$$

$$+\underbrace{\mathbb{E}\left[\theta_r\langle \boldsymbol{q}^r, \boldsymbol{\lambda}^r - \boldsymbol{\lambda}\rangle + \frac{1}{\sigma_r}\mathrm{D}(\boldsymbol{\lambda}, \boldsymbol{\lambda}^r) + \frac{1}{\tau_r}\mathrm{D}(\boldsymbol{x}, \boldsymbol{x}^r) + \frac{\theta_r}{2\alpha_r}\|\boldsymbol{q}^r\|^2\right]}_{V_r} + \frac{\alpha_r\theta_r}{2}\mathbb{E}\left[\|\boldsymbol{\lambda}^{r+1} - \boldsymbol{\lambda}^r\|^2\right]$$

$$+\mathbb{E}\left[\frac{1}{2\alpha_{r+1}}\|\boldsymbol{q}^{r+1}\|^2 + 3L_{\boldsymbol{x}\boldsymbol{x}}\|\boldsymbol{x}^{r+1} - \boldsymbol{x}^r\|^2 + 3L_{\boldsymbol{x}\boldsymbol{x}}\mathcal{E}_r - \frac{1}{2\tau_r}\mathrm{D}(\boldsymbol{x}^{r+1}, \boldsymbol{x}^r) - \frac{1}{\sigma_r}\mathrm{D}(\boldsymbol{\lambda}^{r+1}, \boldsymbol{\lambda}^r)\right] + \frac{2\chi\tau_r}{J}\zeta^2$$

$$\leq -Z_{r+1} + V_r + \frac{\alpha_r\theta_r}{2}\mathbb{E}\left[\|\boldsymbol{\lambda}^{r+1} - \boldsymbol{\lambda}^r\|^2\right] + \frac{L_{\boldsymbol{\lambda}\boldsymbol{x}}^2}{2\alpha_{r+1}}\mathbb{E}\left[\|\boldsymbol{x}^{r+1} - \boldsymbol{x}^r\|^2\right] + 3L_{\boldsymbol{x}\boldsymbol{x}}\left[\|\boldsymbol{x}^{r+1} - \boldsymbol{x}^r\|^2\right]$$

$$+ 3L_{\boldsymbol{x}\boldsymbol{x}}\mathcal{E}_r - \frac{1}{2\tau_r}\mathbb{E}\left[\mathrm{D}(\boldsymbol{x}^{r+1}, \boldsymbol{x}^r)\right] - \frac{1}{\sigma_r}\left[\mathrm{D}(\boldsymbol{\lambda}^{r+1}, \boldsymbol{\lambda}^r)\right] + \frac{2\chi\tau_r}{J}\zeta^2$$

$$= -Z_{r+1} + V_r + \left(\frac{\alpha_r\theta_r}{2} - \frac{1}{2\sigma_r}\right)\mathbb{E}\left[\|\boldsymbol{\lambda}^{r+1} - \boldsymbol{\lambda}^r\|^2\right] + \left(\frac{L_{\boldsymbol{\lambda}\boldsymbol{x}}^2}{2\alpha_{r+1}} + 3L_{\boldsymbol{x}\boldsymbol{x}} - \frac{1}{4\tau_r}\right)\mathbb{E}\left[\|\boldsymbol{x}^{r+1} - \boldsymbol{x}^r\|^2\right]$$

$$+ \frac{2\chi\tau_r}{J}\zeta^2 + \underbrace{3L_{\boldsymbol{x}\boldsymbol{x}}\mathcal{E}_r - \frac{1}{8\tau_r}\mathbb{E}\left[\|\boldsymbol{x}^{r+1} - \boldsymbol{x}^r\|^2\right]}_{\mathcal{T}_4}.$$

$$(B.15)$$

Next, to get the upper bound of $\mathcal{T}_4$, we apply Lemma A.3 to analyze the term $\frac{1}{8\tau_r}\mathbb{E}\left[\|\boldsymbol{x}^{r+1} - \boldsymbol{x}^r\|^2\right]$,

$$\frac{1}{8\tau_r}\mathbb{E}\left[\|\boldsymbol{x}^{r+1} - \boldsymbol{x}^r\|^2\right] \geq -\frac{\tau_r L_{\boldsymbol{x}\boldsymbol{x}}^2}{8}\mathcal{E}_r + \frac{\tau_r}{16}\mathbb{E}\left[\left\|\nabla_{\boldsymbol{x}}\Phi(\boldsymbol{x}^r, \boldsymbol{\lambda}^{r+1})\right\|^2\right]. \tag{B.16}$$

By applying Lemma A.2, we can upper bound the drift error as follows,

$$\mathcal{E}_r \leq \frac{12\tau_r^2}{\eta_g^2}\mathbb{E}\left[\left\|\nabla_{\boldsymbol{x}}\Phi(\boldsymbol{x}^r, \boldsymbol{\lambda}^{r+1})\right\|^2\right] + \left[\frac{8\tau_r^2}{\eta_g^2}(1 + \chi) + \frac{3\tau_r^2}{\eta_g^2 J}\right]\zeta^2, \tag{B.17}$$

Then if we set the effective step size as $\tau_r = O(1/L_{\boldsymbol{x}\boldsymbol{x}})$, the term $\mathcal{T}_4$ can be upper bounded as

$$\mathcal{T}_4 = 3L_{\boldsymbol{x}\boldsymbol{x}}\mathcal{E}_r - \frac{1}{8\tau_r}\mathbb{E}\left[\|\boldsymbol{x}^{r+1} - \boldsymbol{x}^r\|^2\right]$$

$$\leq \left(3L_{\boldsymbol{x}\boldsymbol{x}} + \frac{\tau_r L_{\boldsymbol{x}\boldsymbol{x}}^2}{8}\right)\mathcal{E}_r - \frac{\tau_r}{16}\mathbb{E}\left[\left\|\nabla_{\boldsymbol{x}}\Phi(\boldsymbol{x}^r, \boldsymbol{\lambda}^{r+1})\right\|^2\right]$$

$$\leq \underbrace{\left(\frac{36\tau_r^2 L_{\boldsymbol{x}\boldsymbol{x}}}{\eta_g^2} + \frac{2\tau_r^3 L_{\boldsymbol{x}\boldsymbol{x}}^2}{\eta_g^2} - \frac{\tau_r}{16}\right)}_{\leq 0}\mathbb{E}\left[\left\|\nabla_{\boldsymbol{x}}\Phi(\boldsymbol{x}^r, \boldsymbol{\lambda}^{r+1})\right\|^2\right]$$

$$+ \left(3L_{\boldsymbol{x}\boldsymbol{x}} + \frac{\tau_r L_{\boldsymbol{x}\boldsymbol{x}}^2}{8}\right)\left[\frac{8\tau_r^2}{\eta_g^2}(1 + \chi) + \frac{3\tau_r^2}{\eta_g^2 J}\right]\zeta^2 \tag{B.18}$$

$$\leq 12\tau_r\left(3(1 + \chi) + \frac{1}{J}\right)\zeta^2 \leq C\tau_r\zeta^2,$$

where $C \geq 0$ is a non-negative constant. Then by combining Eq. (B.18) and Eq. (B.15), we have

$$\mathbb{E}\left[L(\boldsymbol{x}^{r+1}, \boldsymbol{\lambda}) - L(\boldsymbol{x}, \boldsymbol{\lambda}^{r+1})\right]$$

$$\leq -Z_{r+1} + V_r + \frac{2\chi\tau_r}{J}\zeta^2 + C\tau_r\zeta^2$$

$$+ \underbrace{\left(\frac{\alpha_r\theta_r}{2} - \frac{1}{2\sigma_r}\right)\mathbb{E}\left[\|\boldsymbol{\lambda}^{r+1} - \boldsymbol{\lambda}^r\|^2\right] + \left(\frac{L_{\boldsymbol{\lambda}\boldsymbol{x}}^2}{2\alpha_{r+1}} + 3L_{\boldsymbol{x}\boldsymbol{x}} - \frac{1}{4\tau_r}\right)\mathbb{E}\left[\|\boldsymbol{x}^{r+1} - \boldsymbol{x}^r\|^2\right]}_{\Delta_r} \tag{B.19}$$

$$= -Z_{r+1} + V_r + \Delta_r + C\tau_r\zeta^2,$$

which completes the proof. $\qquad\square$

### B.1.2 HOW TO SET PARAMETERS IN STRONGLY-CONVEX-CONCAVE (SC-C) SETTING?

Next we study how to set the parameters of Algorithm 1 in the strongly-convex-concave setting.

**Lemma B.2** *In Algorithm 1, if we set the parameters as*

$$\sigma_{-1} = \gamma_0 \bar{\tau}, \quad \sigma_r = \gamma_r \tau_r, \quad \theta_r = \sigma_{r-1}/\sigma_r, \quad \gamma_{r+1} = \gamma_r(1 + \mu_{\boldsymbol{x}} \tau_r), \tag{B.20}$$

*and we set $t_r$ as*

$$t_r = \sigma_r/\sigma_0, \tag{B.21}$$

*then we have*

$$t_r \left( \frac{1}{\tau_r} + \mu_{\boldsymbol{x}} \right) \geq \frac{t_{r+1}}{\tau_{r+1}}, \quad \frac{t_r}{\sigma_r} \geq \frac{t_{r+1}}{\sigma_{r+1}}, \quad \frac{t_r}{t_{r+1}} = \theta_{r+1}. \tag{B.22}$$

*Proof.* Because we have $t_r = \sigma_r/\sigma_0$, then $t_r \left( \frac{1}{\tau_r} + \mu_{\boldsymbol{x}} \right) \geq \frac{t_{r+1}}{\tau_{r+1}}$ can be written as

$$(1 + \tau_r \mu_{\boldsymbol{x}}) \geq \frac{\tau_r}{\tau_{r+1}} \frac{t_{r+1}}{t_r} = \frac{\tau_r}{\tau_{r+1}} \frac{\sigma_{r+1}}{\sigma_r}, \tag{B.23}$$

then due to the updates of $\gamma_r$ ($\gamma_{r+1} = \gamma_r(1 + \tau_r \mu_{\boldsymbol{x}})$) and update of $\sigma_r$ ($\sigma_r = \gamma_r \tau_r$), we have

$$(1 + \tau_r \mu_{\boldsymbol{x}}) = \frac{\gamma_{r+1}}{\gamma_r} = \frac{\sigma_{r+1}}{\tau_{r+1}} \frac{\tau_r}{\sigma_r}, \tag{B.24}$$

therefore, the three inequalities in Eq. (B.22) are satisfied. $\qquad \square$

**Lemma B.3** *For Algorithm 1, we have*

$$\frac{\tau_r}{\sigma_r} = \frac{1}{\gamma_r} = O\left( \frac{1}{r^2} \right), \quad \gamma_r = O\left( r^2 \right), \quad \sigma_r = O(r), \quad \tau_r \sigma_r = \tau_0^2 \gamma_0. \tag{B.25}$$

*Proof.* Since $\tau_{r+1} = \tau_r \sqrt{\gamma_r/\gamma_{r+1}}$, then we have $\tau_r = \tau_0 \sqrt{\gamma_0/\gamma_r}$, then based on the update rule for $\gamma_r$ ($\gamma_{r+1} = \gamma_r(1 + \mu_{\boldsymbol{x}} \tau_r)$), we have

$$\gamma_{r+1} = \gamma_r(1 + \mu_{\boldsymbol{x}} \tau_r) = \gamma_r + \mu_{\boldsymbol{x}} \tau_0 \sqrt{\gamma_0 \gamma_r}. \tag{B.26}$$

Then we apply induction to prove that

$$\gamma_r \geq \frac{\mu_{\boldsymbol{x}}^2 \tau_0^2 \gamma_0}{9} r^2. \tag{B.27}$$

Therefore, for $\sigma_r$, we have

$$\sigma_r = \gamma_r \tau_r = \frac{\gamma_{r+1} - \gamma_r}{\mu_{\boldsymbol{x}}} \geq \tau_0 \sqrt{\gamma_0 \gamma_r} \geq \frac{\mu \tau_0^2 \gamma_0}{3} r, \tag{B.28}$$

and

$$\tau_r \sigma_r = \frac{\sigma_r^2}{\gamma_r} = \frac{(\gamma_{r+1} - \gamma_r)^2}{\mu_{\boldsymbol{x}}^2 \gamma_r} = \tau_0^2 \gamma_0 = \text{constant}, \tag{B.29}$$

furthermore, we have

$$\frac{\tau_r}{\sigma_r} = \frac{1}{\gamma_r} = O\left( \frac{1}{r^2} \right). \tag{B.30}$$

$$\square$$

**Remark B.4** *For the sake of simplicity, we establish the validity of the aforementioned two lemmas by considering the case where the parameter $(1/\tau_r + \mu_{\boldsymbol{x}})$ is used. It is worth noting that in subsequent proofs (Theorem B.6), it suffices to substitute a smaller value of $\mu_{\boldsymbol{x}}$, such as $(1/\tau_r + \mu_{\boldsymbol{x}}/4)$.*

**Proposition B.5** *If we first set $\theta_0 = 1$, then we set $\tau_r, \sigma_r, \theta_r$ such that*

$$\frac{1-\delta}{2\tau_r} \geq 6L_{\boldsymbol{xx}} + \frac{L_{\boldsymbol{\lambda x}}^2}{\alpha_{r+1}}, \quad \frac{1-\delta}{\sigma_r} \geq \theta_r \alpha_r, \tag{B.31}$$

*where $\delta \in (0, 1)$. Then $\Delta_r \leq 0$ for $r = 1, \dots, R$.*

### B.1.3 CONVERGENCE ANALYSIS

Finally, we prove the convergence of Algorithm 1 in the strongly-convex-concave setting.

**Theorem B.6** *Under the assumptions of Theorem 5.1, Algorithm 1 will converge to $\boldsymbol{x}^\star$, and*

$$\mathbb{E}\left[\|\boldsymbol{x}^{R+1} - \boldsymbol{x}^\star\|^2\right] \le \frac{C_1}{R^2}\left[\|\boldsymbol{x}^\star - \boldsymbol{x}^0\|^2 + \|\boldsymbol{\lambda}^0 - \boldsymbol{\lambda}^\star\|^2\right] + \frac{C_2}{R}\zeta^2, \tag{B.32}$$

*where $C_1, C_2 > 0$ are constants.*

*Proof.* For $\theta_r$, we have

$$\theta_{r+1} = \frac{\sigma_r}{\sigma_{r+1}} = \frac{\tau_r\gamma_r}{\tau_{r+1}\gamma_{r+1}} = \sqrt{\frac{\gamma_r}{\gamma_{r+1}}} = \frac{1}{\sqrt{1 + \mu_{\boldsymbol{x}}\tau_r}}, \quad \tau_{r+1} = \tau_r\sqrt{\frac{\gamma_r}{\gamma_{r+1}}} = \theta_{r+1}\tau_r, \tag{B.33}$$

where we apply the fact that $\tau_{r+1} = \tau_r\sqrt{\gamma_r/\gamma_{r+1}}$. Next we set $t_r, \alpha_r$ as

$$t_r = \sigma_r/\sigma_0, \quad \alpha_r = c_\alpha/\sigma_{r-1}, \tag{B.34}$$

where $c_\alpha \in (0, 1)$ is a constant. then Eq. (B.31) can be written as

$$\frac{1-\delta}{\tau_r} \ge 12L_{\boldsymbol{xx}} + \frac{2L_{\boldsymbol{\lambda x}}^2\sigma_r}{c_\alpha}, \quad 1 - (\delta + c_\alpha) \ge 0, \tag{B.35}$$

the second one can be easily satisfied, the first one we apply induction to prove it,

$$\frac{1-\delta}{\tau_{r+1}} = \frac{1-\delta}{\tau_r}\sqrt{\frac{\gamma_{r+1}}{\gamma_r}} \ge \left(12L_{\boldsymbol{xx}} + \frac{2L_{\boldsymbol{\lambda x}}^2\sigma_r}{c_\alpha}\right)\sqrt{\frac{\gamma_{r+1}}{\gamma_r}} \ge \left(12L_{\boldsymbol{xx}} + \frac{2L_{\boldsymbol{\lambda x}}^2\sigma_{r+1}}{c_\alpha}\right), \tag{B.36}$$

where we apply the fact that

$$\gamma_{r+1}/\gamma_r \ge 1, \quad \sigma_{r+1} = \sigma_r\sqrt{\gamma_{k+1}/\gamma_k}. \tag{B.37}$$

Therefore, by Eq. (B.35), we can prove that

$$\Delta_r = \mathbb{E}\left[\left(\frac{\alpha_r\theta_r}{2} - \frac{1}{2\sigma_r}\right)\|\boldsymbol{\lambda}^{r+1} - \boldsymbol{\lambda}^r\|^2 + \left(\frac{L_{\boldsymbol{\lambda x}}^2}{2\alpha_{r+1}} + 3L_{\boldsymbol{xx}} - \frac{1}{4\tau_r}\right)\|\boldsymbol{x}^{r+1} - \boldsymbol{x}^r\|^2\right] \le 0.$$

Meanwhile, given the parameters of Algorithm 1 satisfy Condition 5.1, by Lemma B.2, we have

$$t_{r+1}V_{r+1} \le t_r Z_{r+1}. \tag{B.38}$$

Then, by multiplying Eq. (B.1) and summing up from $r = 0, \cdots, R$, we have

$$\left[\sum_{r=0}^R t_r\right]\mathbb{E}\left[F(\bar{\boldsymbol{x}}^{R+1}, \boldsymbol{\lambda}) - F(\boldsymbol{x}, \bar{\boldsymbol{\lambda}}^{R+1})\right] + \frac{t_R}{\tau_R}\cdot\mathbb{E}\left[\mathrm{D}(\boldsymbol{x}^{R+1}, \boldsymbol{x}^\star)\right]$$
$$\le\frac{t_0}{\tau_0}\mathrm{D}(\boldsymbol{x}^\star, \boldsymbol{x}^0) + \frac{t_0}{\sigma_0}\mathrm{D}(\boldsymbol{\lambda}^\star, \boldsymbol{\lambda}^0) + \sum_{r=0}^R (t_r\cdot C\tau_r\zeta^2), \tag{B.39}$$

where we defined $\bar{\boldsymbol{x}}^{R+1}, \bar{\boldsymbol{\lambda}}^{R+1}$ as

$$\bar{\boldsymbol{x}}^{R+1} = \frac{1}{\sum_{r=0}^R t_r}\sum_{r=0}^R t_r\boldsymbol{x}^r, \quad \bar{\boldsymbol{\lambda}}^{R+1} = \frac{1}{\sum_{r=0}^R t_r}\sum_{r=0}^R t_r\boldsymbol{\lambda}^r, \tag{B.40}$$

because by Lemma B.3, we have

$$\sigma_R/\tau_R = O(R^2), \quad \sum_{r=0}^R t_r = O(R^2), \quad t_R = \sigma_R/\sigma_0, \quad t_r\tau_r = \tau_0^2\gamma_0, \tag{B.41}$$

then we have

$$\mathbb{E}\left[\mathrm{D}(\boldsymbol{x}^{R+1}, \boldsymbol{x}^\star)\right] \le \frac{C_1}{R^2}\left[\frac{t_0}{\tau_0}\mathrm{D}(\boldsymbol{x}^\star, \boldsymbol{x}^0) + \frac{t_0}{\sigma_0}\mathrm{D}(\boldsymbol{\lambda}^\star, \boldsymbol{\lambda}^0)\right] + \frac{C_2}{R}\zeta^2, \tag{B.42}$$

where we apply the fact that

$$F(\bar{\boldsymbol{x}}^{R+1}, \boldsymbol{\lambda}^\star) - F(\boldsymbol{x}^\star, \bar{\boldsymbol{\lambda}}^{R+1}) \ge 0. \tag{B.43}$$

This completes our proof. $\qquad\square$

## B.2 PROOFS – STRONGLY-CONVEX-STRONGLY-CONCAVE (SC-SC) SETTING

In this subsection, we first present the technical lemma in Section B.2.1. Next, we analyze how to set the step size related parameters in Section B.2.2. Finally, we prove Theorem 5.5 in Section B.2.3.

### B.2.1 TECHNICAL LEMMAS

**Lemma B.7** *If we set the step size in Alg 1 as $\tau \cdot L_{\boldsymbol{xx}} \leq 1$, then for any $\boldsymbol{x}, \boldsymbol{\lambda}$ we have*

$$\mathbb{E}\left[F(\boldsymbol{x}^{r+1}, \boldsymbol{\lambda}) - F(\boldsymbol{x}, \boldsymbol{\lambda}^{r+1})\right] \leq -Z_{r+1} + V_r + \Delta_r + C\tau\zeta^2, \tag{B.44}$$

*where $Z_r, V_r, \Delta_r$ are defined as*

$$Z_{r+1} = \mathbb{E}\left[\langle \boldsymbol{q}^{r+1}, \boldsymbol{\lambda}^{r+1} - \boldsymbol{\lambda}\rangle + \left(\frac{1}{2\sigma} + \frac{\mu_{\boldsymbol{\lambda}}}{2}\right)\|\boldsymbol{\lambda}^{r+1} - \boldsymbol{\lambda}\|^2 + \left(\frac{1}{2\tau} + \frac{\mu_{\boldsymbol{x}}}{8}\right)\|\boldsymbol{x}^{r+1} - \boldsymbol{x}\|^2\right],$$

$$V_r = \mathbb{E}\left[\theta^r\langle \boldsymbol{q}^r, \boldsymbol{\lambda}^r - \boldsymbol{\lambda}\rangle + \frac{1}{2\sigma}\|\boldsymbol{\lambda}^r - \boldsymbol{\lambda}\|^2 + \frac{1}{2\tau}\|\boldsymbol{x}^r - \boldsymbol{x}\|^2\right], \tag{B.45}$$

$$\Delta_r = \mathbb{E}\left[\left(\frac{\theta L_{\boldsymbol{\lambda x}}}{2\pi} - \frac{1}{2\sigma}\right)\|\boldsymbol{\lambda}^{r+1} - \boldsymbol{\lambda}^r\|^2 + \left(\frac{\pi\theta L_{\boldsymbol{\lambda x}}}{2} + 3L_{\boldsymbol{xx}} - \frac{1}{4\tau}\right)\|\boldsymbol{x}^{r+1} - \boldsymbol{x}^r\|^2\right],$$

*where $\pi > 0$ is a parameter and $C \geq 0$ is a constant.*

*Proof.* Most of the steps are the same as in the Lemma B.1. To start with, based on the condition that $\psi(\boldsymbol{\lambda})$ is strongly convex in $\boldsymbol{\lambda}$, we apply Lemma A.4,

$$\psi(\boldsymbol{\lambda}^{r+1}) - \langle \boldsymbol{s}^r, \boldsymbol{\lambda}^{r+1} - \boldsymbol{\lambda}\rangle$$
$$\leq \psi(\boldsymbol{\lambda}^r) + \frac{1}{\sigma}\left[D(\boldsymbol{\lambda}, \boldsymbol{\lambda}^r) - D(\boldsymbol{\lambda}, \boldsymbol{\lambda}^{r+1}) - D(\boldsymbol{\lambda}^{r+1}, \boldsymbol{\lambda}^r)\right] - \frac{\mu_{\boldsymbol{\lambda}}}{2}\|\boldsymbol{\lambda}^{r+1} - \boldsymbol{\lambda}\|^2. \tag{B.46}$$

Next, we change the way we upper bound $\theta\langle \boldsymbol{q}^r, \boldsymbol{\lambda}^{r+1} - \boldsymbol{\lambda}^r\rangle$ in the strongly-convex-concave setting, and we upper bound this term as follows,

$$\begin{aligned}\theta\langle \boldsymbol{q}^r, \boldsymbol{\lambda}^{r+1} - \boldsymbol{\lambda}^r\rangle &= \theta\langle \nabla_{\boldsymbol{\lambda}}\Phi(\boldsymbol{x}^r, \boldsymbol{\lambda}^r) - \nabla_{\boldsymbol{\lambda}}\Phi(\boldsymbol{x}^{r-1}, \boldsymbol{\lambda}^{r-1}), \boldsymbol{\lambda}^{r+1} - \boldsymbol{\lambda}^r\rangle \\ &= \theta\langle \nabla_{\boldsymbol{\lambda}}\Phi(\boldsymbol{x}^r, \boldsymbol{\lambda}^r) - \nabla_{\boldsymbol{\lambda}}\Phi(\boldsymbol{x}^{r-1}, \boldsymbol{\lambda}^r), \boldsymbol{\lambda}^{r+1} - \boldsymbol{\lambda}^r\rangle \\ &\leq \theta\|\nabla_{\boldsymbol{\lambda}}\Phi(\boldsymbol{x}^r, \boldsymbol{\lambda}^r) - \nabla_{\boldsymbol{\lambda}}\Phi(\boldsymbol{x}^{r-1}, \boldsymbol{\lambda}^r)\|\|\boldsymbol{\lambda}^{r+1} - \boldsymbol{\lambda}^r\| \\ &\leq \frac{\pi\theta L_{\boldsymbol{\lambda x}}}{2}\|\boldsymbol{x}^r - \boldsymbol{x}^{r-1}\|^2 + \frac{\theta L_{\boldsymbol{\lambda x}}}{2\pi}\|\boldsymbol{\lambda}^{r+1} - \boldsymbol{\lambda}^r\|^2, \end{aligned} \tag{B.47}$$

where $\pi > 0$ is a constant. Then we have

$$F(\boldsymbol{x}^{r+1}, \boldsymbol{\lambda}) - F(\boldsymbol{x}, \boldsymbol{\lambda}^{r+1})$$

$$\leq -\underbrace{\mathbb{E}\left[\langle \boldsymbol{q}^{r+1}, \boldsymbol{\lambda}^{r+1} - \boldsymbol{\lambda}\rangle + \frac{1}{\tau}D(\boldsymbol{x}, \boldsymbol{x}^{r+1}) + \frac{\mu_{\boldsymbol{x}}}{8}\|\boldsymbol{x}^{r+1} - \boldsymbol{x}\|^2 + \frac{1}{\sigma}D(\boldsymbol{\lambda}, \boldsymbol{\lambda}^{r+1}) + \frac{\mu_{\boldsymbol{\lambda}}}{2}\|\boldsymbol{\lambda}^{r+1} - \boldsymbol{\lambda}\|^2\right]}_{Z_{r+1}}$$

$$+ \underbrace{\mathbb{E}\left[\theta\langle \boldsymbol{q}^r, \boldsymbol{\lambda}^r - \boldsymbol{\lambda}\rangle + \frac{1}{\sigma}D(\boldsymbol{\lambda}, \boldsymbol{\lambda}^r) + \frac{1}{\tau}D(\boldsymbol{x}, \boldsymbol{x}^r)\right]}_{V_r} + \frac{\pi\theta L_{\boldsymbol{\lambda x}}}{2}\mathbb{E}\left[\|\boldsymbol{x}^r - \boldsymbol{x}^{r-1}\|^2\right] + \frac{\theta L_{\boldsymbol{\lambda x}}}{2\pi}\mathbb{E}\left[\|\boldsymbol{\lambda}^{r+1} - \boldsymbol{\lambda}^r\|^2\right]$$

$$+ 3L_{\boldsymbol{xx}}\mathbb{E}\left[\|\boldsymbol{x}^{r+1} - \boldsymbol{x}^r\|^2\right] + 3L_{\boldsymbol{xx}}\mathcal{E}_r - \frac{1}{2\tau}\mathbb{E}\left[D(\boldsymbol{x}^{r+1}, \boldsymbol{x}^r)\right] - \frac{1}{\sigma}\mathbb{E}\left[D(\boldsymbol{\lambda}^{r+1}, \boldsymbol{\lambda}^r)\right] + \frac{2\chi\tau}{J}\zeta^2$$

$$= -Z_{r+1} + V_r + \left(\frac{\theta L_{\boldsymbol{\lambda x}}}{2\pi} - \frac{1}{2\sigma}\right)\mathbb{E}\left[\|\boldsymbol{\lambda}^{r+1} - \boldsymbol{\lambda}^r\|^2\right] + \left(\frac{\pi\theta L_{\boldsymbol{\lambda x}}}{2} + 3L_{\boldsymbol{xx}} - \frac{1}{4\tau}\right)\mathbb{E}\left[\|\boldsymbol{x}^{r+1} - \boldsymbol{x}^r\|^2\right]$$

$$+ 3L_{\boldsymbol{xx}}\mathcal{E}_r - \frac{1}{8\tau}\mathbb{E}\left[\|\boldsymbol{x}^{r+1} - \boldsymbol{x}^r\|^2\right] + \frac{2\chi\tau}{J}\zeta^2$$

$$\geq -Z_{r+1} + V_r + \underbrace{\left(\frac{\theta L_{\boldsymbol{\lambda x}}}{2\pi} - \frac{1}{2\sigma}\right)\mathbb{E}\left[\|\boldsymbol{\lambda}^{r+1} - \boldsymbol{\lambda}^r\|^2\right] + \left(\frac{\pi\theta L_{\boldsymbol{\lambda x}}}{2} + 3L_{\boldsymbol{xx}} - \frac{1}{4\tau}\right)\mathbb{E}\left[\|\boldsymbol{x}^{r+1} - \boldsymbol{x}^r\|^2\right]}_{\Delta_r}$$

$$+ C\tau\zeta^2,$$

where the last inequality is because Eq. (B.18). This completes our proof. $\square$

### B.2.2 HOW TO SET PARAMETERS IN STRONGLY-CONVEX-STRONGLY-CONCAVE (SC-SC) SETTING?

**Lemma B.8** *For Algorithm 1, if we set the parameters as*

$$\mu_{\boldsymbol{x}}\tau = O\left(\frac{1-\theta}{\theta}\right), \quad \mu_{\boldsymbol{\lambda}}\sigma = O\left(\frac{1-\theta}{\theta}\right), \quad \frac{1}{1-\theta} = O\left(\frac{L_{\boldsymbol{xx}}}{\mu_{\boldsymbol{x}}} + \sqrt{\frac{L_{\boldsymbol{\lambda x}}^2}{\mu_{\boldsymbol{x}}\mu_{\boldsymbol{\lambda}}}}\right), \qquad \text{(B.48)}$$

*then we have*

$$\frac{1}{2\tau} + \frac{\mu_{\boldsymbol{x}}}{8} \geq \frac{1}{2\tau\theta}, \quad \frac{1}{2\sigma} + \frac{\mu_{\boldsymbol{\lambda}}}{2} \geq \frac{1}{2\sigma\theta}, \quad \frac{1}{\tau} \geq 12L_{\boldsymbol{xx}} + 2\pi\theta L_{\boldsymbol{\lambda x}}, \quad \frac{1}{\sigma} \geq \frac{\theta L_{\boldsymbol{\lambda x}}}{\pi}. \qquad \text{(B.49)}$$

*Proof.* The conditions in Eq. (B.49) can be reformulated as follows,

$$
\begin{aligned}
\frac{1}{2\tau} + \frac{\mu_{\boldsymbol{x}}}{8} \geq \frac{1}{2\tau\theta} &\quad\Leftrightarrow\quad \mu_{\boldsymbol{x}}\tau \geq 4\frac{1-\theta}{\theta}, \\
\frac{1}{2\sigma} + \frac{\mu_{\boldsymbol{\lambda}}}{2} \geq \frac{1}{2\sigma\theta} &\quad\Leftrightarrow\quad \mu_{\boldsymbol{\lambda}}\sigma \geq \frac{1-\theta}{\theta}, \\
\frac{1}{\tau} \geq 12L_{\boldsymbol{xx}} + 2\pi\theta L_{\boldsymbol{\lambda x}} &\quad\Leftarrow\quad \frac{1}{\tau} \geq 12L_{\boldsymbol{xx}} + 2\pi L_{\boldsymbol{\lambda x}}, \\
\frac{1}{\sigma} \geq \frac{\theta L_{\boldsymbol{\lambda x}}}{\pi} &\quad\Leftarrow\quad \frac{c}{\sigma} \geq \frac{\theta L_{\boldsymbol{\lambda x}}}{\pi},
\end{aligned}
\qquad \text{(B.50)}
$$

where $c \in (0, 1]$.

Next we study how to set $\{\tau, \sigma, \theta\}$ such that Eq. (B.50) holds, we could set

$$
\begin{aligned}
\tau \geq \frac{4}{\mu_{\boldsymbol{x}}}\frac{1-\theta}{\theta}, &\quad \sigma \geq \frac{1}{\mu_{\boldsymbol{\lambda}}}\frac{1-\theta}{\theta}, \\
\pi &= \frac{\theta\sigma L_{\boldsymbol{\lambda x}}}{c}, \\
\frac{\mu_{\boldsymbol{x}}\theta}{4(1-\theta)} - 12L_{\boldsymbol{xx}} &\geq \frac{2\theta\sigma L_{\boldsymbol{\lambda x}}^2}{c} \geq (1-\theta)\frac{2L_{\boldsymbol{\lambda x}}^2}{c\mu_{\boldsymbol{\lambda}}},
\end{aligned}
\qquad \text{(B.51)}
$$

therefore, once $\theta$ satisfy the following condition

$$\frac{\mu_{\boldsymbol{x}}\theta}{4(1-\theta)} - 12L_{\boldsymbol{xx}} \geq (1-\theta)\frac{2L_{\boldsymbol{\lambda x}}^2}{c\mu_{\boldsymbol{\lambda}}}, \qquad \text{(B.52)}$$

and then we can set $\tau$ and $\sigma$ based on the value of $\theta$ according to Eq. (B.51). Then if we let

$$\omega = \frac{1}{1-\theta}, \qquad \text{(B.53)}$$

therefore, based on Eq. (B.52), by setting $c = 1$, we have

$$
\begin{aligned}
&\frac{\omega-1}{\omega}\frac{\omega\mu_{\boldsymbol{x}}}{4} - 12L_{\boldsymbol{xx}} \geq \frac{1}{\omega}\frac{2L_{\boldsymbol{\lambda x}}^2}{\mu_{\boldsymbol{\lambda}}}, \\
\Leftrightarrow\quad &\mu_{\boldsymbol{x}}\mu_{\boldsymbol{\lambda}}\omega^2 - (\mu_{\boldsymbol{x}}\mu_{\boldsymbol{\lambda}} + 48\mu_{\boldsymbol{\lambda}}L_{\boldsymbol{xx}})\omega - 8L_{\boldsymbol{\lambda x}}^2 \geq 0, \\
\Leftarrow\quad &\omega = C_\omega \frac{(\mu_{\boldsymbol{x}}\mu_{\boldsymbol{\lambda}} + 48\mu_{\boldsymbol{\lambda}}L_{\boldsymbol{xx}}) + \sqrt{(\mu_{\boldsymbol{x}}\mu_{\boldsymbol{\lambda}} + 48\mu_{\boldsymbol{\lambda}}L_{\boldsymbol{xx}})^2 + 32\mu_{\boldsymbol{x}}\mu_{\boldsymbol{\lambda}}L_{\boldsymbol{\lambda x}}^2}}{2\mu_{\boldsymbol{x}}\mu_{\boldsymbol{\lambda}}}, \\
\Leftrightarrow\quad &\omega = C_\omega\left(\frac{1}{2} + \frac{24L_{\boldsymbol{xx}}}{\mu_{\boldsymbol{x}}} + \sqrt{\left(\frac{1}{2} + \frac{24L_{\boldsymbol{xx}}}{\mu_{\boldsymbol{x}}}\right)^2 + \frac{16L_{\boldsymbol{\lambda x}}^2}{\mu_{\boldsymbol{x}}\mu_{\boldsymbol{\lambda}}}}\right), \\
\Leftrightarrow\quad &\omega = O\left(\frac{L_{\boldsymbol{xx}}}{\mu_{\boldsymbol{x}}} + \sqrt{\frac{L_{\boldsymbol{\lambda x}}^2}{\mu_{\boldsymbol{x}}\mu_{\boldsymbol{\lambda}}}}\right),
\end{aligned}
\qquad \text{(B.54)}
$$

where $C_\omega \geq 1$ is a constant. This completes our proof.

$\square$

### B.2.3 CONVERGENCE ANALYSIS

**Theorem B.9** *Under the assumptions in Theorem 5.5, Algorithm 1 will converge to $\boldsymbol{x}^\star$, and*

$$\mathbb{E}\left[\|\boldsymbol{x}^r - \boldsymbol{x}^\star\|^2\right] \leq C_1 \theta^R \left[\|\boldsymbol{x}^0 - \boldsymbol{x}^\star\|^2 + \|\boldsymbol{\lambda}^0 - \boldsymbol{\lambda}^\star\|^2\right] + C_2(1-\theta)\frac{\zeta^2}{\mu_{\boldsymbol{x}}^2}, \tag{B.55}$$

*where $C_1, C_2 \geq 0$ are non-negative constants.*

*Proof.* The last two conditions in Eq. (B.49) ensure

$$\Delta_r = \mathbb{E}\left[\left(\frac{\theta L_{\boldsymbol{\lambda x}}}{2\pi} - \frac{1}{2\sigma}\right)\|\boldsymbol{\lambda}^{r+1} - \boldsymbol{\lambda}^r\|^2 + \left(\frac{\pi\theta L_{\boldsymbol{\lambda x}}}{2} + 3L_{\boldsymbol{xx}} - \frac{1}{4\tau}\right)\|\boldsymbol{x}^{r+1} - \boldsymbol{x}^r\|^2\right] \leq 0,$$

for $r = 0, \ldots, R$. The first two conditions in Eq. (B.49) ensure

$$Z_{r+1} \geq \frac{1}{\theta}V_{r+1}.$$

Therefore, by applying Lemma B.7, we have

$$\mathbb{E}\left[F(\boldsymbol{x}^{r+1}, \boldsymbol{\lambda}) - F(\boldsymbol{x}, \boldsymbol{\lambda}^{r+1})\right] + \frac{1}{\theta}V_{r+1} \leq V_r + \Delta_r + C\tau\zeta^2, \tag{B.56}$$

Then we plug $\boldsymbol{x} = \boldsymbol{x}^\star, \boldsymbol{\lambda} = \boldsymbol{\lambda}^\star$ in Eq. (B.44), and we have $F(\boldsymbol{x}^{r+1}, \boldsymbol{\lambda}^\star) - F(\boldsymbol{x}^\star, \boldsymbol{\lambda}^{r+1}) \geq 0$, then we have

$$V_{r+1} \leq \theta V_r + \theta\Delta_r + C\theta\tau\zeta^2, \tag{B.57}$$

therefore, we can derive that

$$V_R \leq \theta^R V_0 + \theta\Delta_R + \frac{C\tau\theta\zeta^2}{1-\theta}, \tag{B.58}$$

meanwhile, we can set the parameters $\{\tau, \sigma, \theta\}$ (according to Eq. (B.50)) such that

$$\mathbb{E}\left[\|\boldsymbol{x}^r - \boldsymbol{x}\|^2\right] \leq 4\tau\theta^R V_0 + \frac{C\tau^2\theta\zeta^2}{1-\theta} \tag{B.59}$$

since $\tau = 2(1-\theta)/(\theta\mu_{\boldsymbol{x}})$,

$$\mathbb{E}\left[\|\boldsymbol{x}^r - \boldsymbol{x}\|^2\right] \leq 4\tau\theta^R V_0 + \frac{4C(1-\theta)\zeta^2}{\theta\mu_{\boldsymbol{x}}^2} \tag{B.60}$$

We need to run at least $N_\varepsilon$ rounds such that $4\tau\theta^R V_0 + \frac{4C(1-\theta)\zeta^2}{\theta\mu_{\boldsymbol{x}}^2} \leq 2\varepsilon$.

Suppose $N_\varepsilon$ satisfies

$$N_\varepsilon = O\left(\ln\left(\frac{V_0}{\varepsilon}\right) \bigg/ \ln\left(\frac{1}{\theta}\right)\right), \tag{B.61}$$

then we have $4\tau\theta^R V_0 \leq \varepsilon$. Because $\ln(1/\theta)$ is convex in $\theta \in \mathbb{R}_+$, then we have

$$\ln\left(\frac{1}{\theta}\right) \leq \frac{1}{1-\theta}, \quad \theta \in (0,1), \tag{B.62}$$

therefore, to get an upper bound for $N_\varepsilon$, we only need to get the upper bound for $\frac{1}{1-\theta}$. Then if we set $\omega = \frac{1}{1-\theta}$, then based on Eq. (B.54), we have

$$\omega = O\left(\frac{L_{\boldsymbol{xx}}}{\mu_{\boldsymbol{x}}} + \sqrt{\frac{L_{\boldsymbol{\lambda x}}^2}{\mu_{\boldsymbol{x}}\mu_{\boldsymbol{\lambda}}}}\right), \tag{B.63}$$

meanwhile, we need to ensure $\frac{4C(1-\theta)\zeta^2}{\theta\mu_{\boldsymbol{x}}^2}$ is small, i.e.,

$$\frac{4C(1-\theta)\zeta^2}{\theta\mu_{\boldsymbol{x}}^2} = \varepsilon \quad \Leftrightarrow \quad \frac{1}{1-\theta} = \frac{4C\zeta^2}{\theta\mu_{\boldsymbol{x}}^2\varepsilon}. \tag{B.64}$$

therefore, in ensure $\mathbb{E}\left[\|\boldsymbol{x}^r - \boldsymbol{x}\|^2\right] \leq 2\varepsilon$, the number of communication rounds satisfies

$$N_\varepsilon = \widetilde{O}\left(\frac{L_{\boldsymbol{xx}}}{\mu_{\boldsymbol{x}}} + \sqrt{\frac{L_{\boldsymbol{\lambda x}}^2}{\mu_{\boldsymbol{x}}\mu_{\boldsymbol{\lambda}}}} + \frac{\zeta^2}{\mu_{\boldsymbol{x}}^2\varepsilon}\right), \tag{B.65}$$

which completes our proof. $\qquad\square$

# C ADDITIONAL IMPLEMENTATION DETAILS AND EXPERIMENTAL RESULTS

In this section, we provide further details for algorithm implementations (Section C.1) as well as additional experimental results – trade-off between worst-20% and average accuracy (Section C.2), convergence performance on synthetic datasets (Section C.3), and comparison with existing methods (Section C.4).

## C.1 ADDITIONAL EXPERIMENTAL DETAILS

In order to enhance the performance of baseline methods, we incorporate local steps into the AFL (Mohri et al., 2019) method. We find that employing local steps yields significantly better performance compared to taking a single gradient step. To ensure a fair comparison, we employ identical feature extraction procedures across all methods. Following the setup outlined in Yu et al. (2022), we first compute the empirical neural tangent kernel (eNTK) representations of the input samples. Then, we randomly select 50,000 features from the eNTK representation through subsampling. For the (local) objective function, we utilize the mean squared error (MSE) loss, which has been used for classification tasks as described in Yu et al. (2022). To calculate the average accuracy, we begin by computing the test accuracy of each client. Then, we compute the average accuracy by averaging the results from all clients.

## C.2 TRADE-OFF BETWEEN WORST-20% ACCURACY AND AVERAGE ACCURACY

We present the trade-off between worst-20% accuracy and average/best-20% accuracy through a scatter plot, as illustrated in Figure 4. We consider the TinyImageNet dataset with the Non-i.i.d. degree parameter $\alpha = 0.01$. Our proposed algorithm, as illustrated in Figure 4, showcases a compelling trade-off between accuracy in the worst-20% and the average/best-20% scenarios.

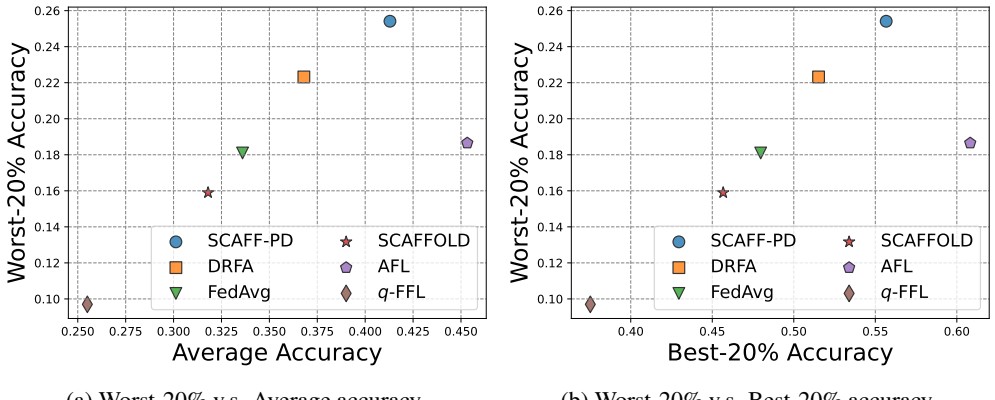

(a) Worst-20% v.s. Average accuracy.          (b) Worst-20% v.s. Best-20% accuracy.

Figure 4: Compare the average/worst-20%/best-20% accuracy of different algorithms on TinyImageNet with $\alpha = 0.01$.

### C.3 Additional Experiments on Synthetic Datasets

We vary the level of data heterogeneity by changing the parameter $\sigma$ from 0.01 to 0.1, where $\sigma$ is used for generating $\delta_i^{\boldsymbol{x}}$ ($\delta_i^{\boldsymbol{x}} \sim \mathcal{N}(\mathbf{0}, \sigma^2 \boldsymbol{I}_{d \times d})$). Figure 5a illustrates the fast convergence of SCAFF-PD to the optimal solution across various data heterogeneity settings. We also explore the effect of varying the number of local steps. Figure 5b demonstrates that increasing the number of local steps results in faster convergence towards the optimal solution.

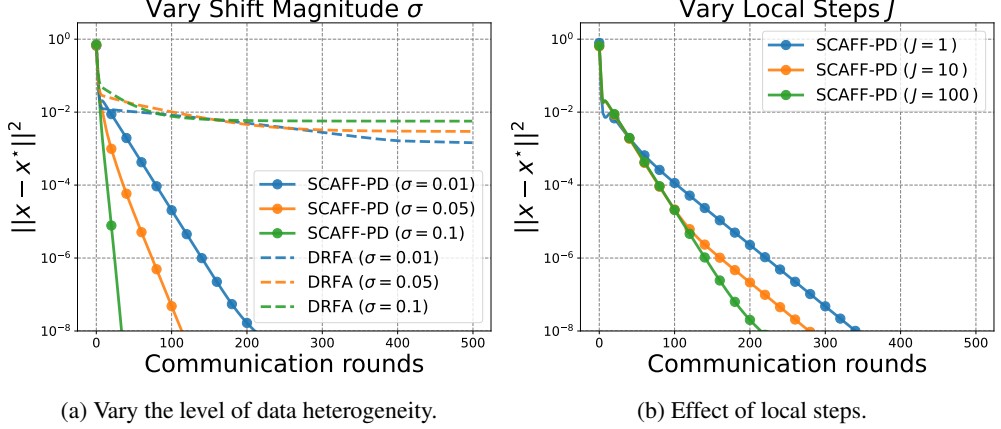

(a) Vary the level of data heterogeneity.

(b) Effect of local steps.

Figure 5: (**left**) Compare SCAFF-PD and DRFA under different levels of data heterogeneity. (**right**) Study the effect of local steps for our proposed algorithm on the synthetic dataset.

### C.4 Additional Experiments on Comparison with Existing Methods

**More clients.** On the CIFAR100 dataset, we conduct a comparison of different algorithms in the 50 clients setting, following the configuration outlined in Table 1. The summarized results are presented in Table 2. Consistent with our previous findings, SCAFF-PD exhibits superior robustness when compared to existing methods.

Table 2: The average and worst-20% top-1 accuracy of our algorithm (SCAFF-PD) vs. state-of-the-art federated learning algorithms evaluated on CIFAR100 with 50 clients. The highest top-1 accuracy in each setting is highlighted in **bold**.

| Datasets | Methods | Non-i.i.d. degree | |
|---|---|---|---|
| | | $\alpha = 0.01$ | |
| | | average | worst-20% |
| | FedAvg | 45.45 | 20.64 |
| | SCAFFOLD | 43.73 | 18.33 |
| | $q$-FFL | 33.42 | 8.13 |
| CIFAR-100 | AFL | 49.93 | 31.87 |
| | DRFA | **51.07** | 31.23 |
| | *SCAFF-PD* | 50.43 | **33.03** |

**Additional dataset.** We consider another dataset – CIFAR10 dataset, the setup mostly follows the configuration outlined in Table 1. We summarize the results in Table 3. We observe that SCAFF-PD outperforms existing methods.

**Additional baselines.** In addition to the baseline methods listed in Table 1, we include $\Delta$-FL (Pillutla et al., 2021) and FedProx (Li et al., 2020b) in our evaluation. We adopt a similar setup as presented in Table 1 to assess the performance of these two methods. The summarized results are presented in Table 4, indicating that our proposed algorithm surpasses both $\Delta$-FL and FedProx in terms of worst-20% accuracy and average accuracy.

Table 3: The average and worst-20% top-1 accuracy of our algorithm (SCAFF-PD) vs. state-of-the-art federated learning algorithms evaluated on CIFAR10 with 20 clients and $\alpha = 0.05$. The highest top-1 accuracy in each setting is highlighted in **bold**.

| Datasets | Methods | Non-i.i.d. degree | |
|---|---|---|---|
| | | $\alpha = 0.05$ | |
| | | average | worst-20% |
| | FedAvg | 77.42 | 60.63 |
| | SCAFFOLD | 77.75 | 62.89 |
| | *q*-FFL | 68.52 | 41.26 |
| CIFAR-100 | AFL | 78.89 | 65.07 |
| | DRFA | 79.04 | 65.02 |
| | *SCAFF-PD* | **79.71** | **69.59** |

Table 4: The average and worst-20% top-1 accuracy of our algorithm (SCAFF-PD) vs. state-of-the-art federated learning algorithms evaluated on CIFAR100 and Tiny-ImageNet. The highest top-1 accuracy in each setting is highlighted in **bold**.

| Datasets | Methods | Non-i.i.d. degree | | | | | |
|---|---|---|---|---|---|---|---|
| | | $\alpha = 0.01$ | | $\alpha = 0.05$ | | $\alpha = 0.1$ | |
| | | average | worst-20% | average | worst-20% | average | worst-20% |
| | FedProx | 38.76 | 15.58 | 35.91 | 24.57 | 36.49 | 26.45 |
| CIFAR-10 | $\Delta$-FL | 30.09 | 7.26 | 33.18 | 15.82 | 31.69 | 16.63 |
| | *SCAFF-PD* | **49.03** | **29.30** | **42.06** | **28.37** | **43.69** | **32.77** |
| | | average | worst-20% | average | worst-20% | average | worst-20% |
| | FedProx | 33.65 | 18.09 | 31.52 | 23.62 | 34.98 | 27.59 |
| TinyImageNet | $\Delta$-FL | 29.06 | 11.94 | 36.77 | 22.24 | 36.47 | 20.13 |
| | *SCAFF-PD* | **41.26** | **25.32** | **39.32** | **30.27** | **41.23** | **29.78** |