# OpenReview forum: "Scaff-PD: Communication Efficient Fair and Robust Federated Learning"
_ICLR.cc/2024/Conference — Submitted to ICLR 2024_

### Official Review · Reviewer_b279 · 2023-10-30

**Soundness:** 3 good
**Presentation:** 3 good
**Contribution:** 2 fair
**Rating:** 5
**Confidence:** 4

**Summary:**

This work studied a distributionally robust federated learning problem, where the problem tries to improve the fairness among clients by optimizing a class of distributionally robust objective functions plus a regularizer that aims to keep the weights $\lambda$ not far from the average weights. The resulting problem now becomes a min-max problem, where the max is take over weights within a prior constraint set $\Gamma$ and the min is taken over the model parameters $x$ to minimize the weighted loss across all clients. The regularizer is chosen to be the penalty from Levy et al., 2020. The authors propose an accelerated primal-duel federated method using some tools from the variance-reduction idea in SCAFFOLD as well as an extrapolation step for acceleration. In theory, the authors characterize the convergence rate for the proposed method for the strongly-convex-concave and strongly-convex-strongly-concave geometry, which matches with centralized version under certain conditions. Experiments are provided to demonstrate the effectiveness of the method.

**Strengths:**

1.	The work is well written. Distributionally robust objectives in FL are important to achieve certain-level fairness by optimizing the worst-case distribution over clients rather than their simple average. The formulated problem is easy to following and well structured.

2.	The proposed algorithms incorporate several ideas in the primal-duel designs and the theory shows some improvements.

**Weaknesses:**

1.	The formulation seems not to be quite new given existing studies. Although the authors show that their formulation is a generalization of existing ones, directly applying the distributionally robust objective (i.e., client weights optimized over a set of candidate distribution) is not new. The $\Phi$ regularizer seems not to add too much new things into the formulation as well. In sum, although the formulation is general, the idea and individual components are not new.

2.	The algorithms do not incorporate many new stuffs. For example, the SCAFFORD bias correction, the extrapolated step for min-max acceleration have been well studied. The proposed algorithm seems to apply to the general federated min-max problem. However, since the DRO-FL problem has its special structure, i.e., $\lambda$ is linear in the first part of the total objective and the regularizer may still have some benign and dedicated structure in terms of $\lambda$. It may be more interesting to explore such architecture to get better results rather than in the worst-case setting of federated minmax case. In sum, the developed algorithms are not existing enough.

3.	Theory is not strong. The analysis is only conducted for strongly-convex-(strongly)-concave settings, while most practical examples work under the nonconvex setting. In addition, there are quite a few works on general federated nonconvex-concave/PL/strong-concave setting (see https://arxiv.org/abs/2302.04249 and related works therein). It would be great to discuss why only strongly-convex-(strongly)-concave settings are studied here. Again, the analysis is mainly developed for general case without taking the DRO min-max structure into account.

4.	Experiments can be made more convincing. The improvements in Table 1 are not that significant. In some cases, the worst-20% results are improved with too much loss in average accuracy. For example, for $\alpha=0.1$, worst-20% increases from 29.5 to 29.78, but the average accuracy drops from 46.11 to 41.23.

**Questions:**

Overall, I think this is a very important problem but the algorithms, analysis and experiments are not novel and convincing enough. Thus, I give a weak reject but open to increase given the feedback and others’ comments. My questions can be found in the weakness part.

---

> ### Author Response · Authors · 2023-11-22
> **Author response (part 1)**
>
> >**Q1**: The formulation seems not to be quite new given existing studies. Although the authors show that their formulation is a generalization of existing ones, directly applying the distributionally robust objective (i.e., client weights optimized over a set of candidate distribution) is not new. The $\Phi$ regularizer seems not to add too much new things into the formulation as well. In sum, although the formulation is general, the idea and individual components are not new.
>
> **A1**: Thank you for your suggestions on clarifying the contributions of our work. We agree that the formulation is not new, and we are leveraging formulations developed in the DRO literature. Instead, **our work is trying to tackle the optimization problem of how to solve the DRO problem more efficiently in federated learning**. In below, we would like to reiterate our key theoretical contributions of this work:
> - New provable algorithms for solving DRO in federated learning. In this work, we (1). design new algorithm components to perform gradient correction; (2). analyze and control the drift error on each device such that the proposed algorithm converges to the optimal solution; and (3). The resulting algorithms – SCAFF-PD – are guaranteed to converge to the optimal saddle point solution of DRO problems in federated learning.
> - Faster convergence rates in both strongly-convex-strongly-concave setting and strongly-convex-concave setting. Our analysis shows that SCAFF-PD enjoys faster convergence rates compared to existing algorithms [BDK+2022, HTF+2021] in the federate learning literature, for both strongly-convex-strongly-concave setting and strongly-convex-concave setting.
>
>
> >**Q2**: The algorithms do not incorporate many new stuffs. For example, the SCAFFORD bias correction, the extrapolated step for min-max acceleration have been well studied. The proposed algorithm seems to apply to the general federated min-max problem. However, since the DRO-FL problem has its special structure, i.e., $\lambda$ is linear in the first part of the total objective and the regularizer may still have some benign and dedicated structure in terms of $\lambda$. It may be more interesting to explore such architecture to get better results rather than in the worst-case setting of federated minmax case. In sum, the developed algorithms are not existing enough.
>
> **A2**: There seems to be a misunderstanding. **Our proposed algorithm SCAFF-PD is exploring and leveraging the special structure** —$\lambda$ is linear in the first part of the total objective— of the DRO problem. For example, our proposed algorithm **cannot** be directly applied to solve generic min-max optimization problems in federated learning. The update for the dual variable (Eq. (4.1)) relies on the special structure (the coupling term is linear in $\lambda$). Furthermore, we were able to achieve the accelerated convergence rates that are faster than all existing algorithms because we leveraged the special structure of the DRO, **whereas most existing min-max federated learning algorithms [BDK+2022, HTF+2021] are designed for solving generic min-max optimization problems and do not achieve the accelerated rate developed in this work**.
>
>
> [BDK+2022] A. Beznosikov, P. Dvurechenskii, A. Koloskova, V. Samokhin, S. U. Stich, and A. Gasnikov. Decentralized local stochastic extra-gradient for variational inequalities. Advances in Neural Information Processing Systems, 35:38116–38133, 2022.
>
> [HTF+2021] C. Hou, K. K. Thekumparampil, G. Fanti, and S. Oh. Efficient algorithms for federated saddle point optimization. arXiv preprint arXiv:2102.06333, 2021.

---

> ### Author Response · Authors · 2023-11-23
> **Author response (part 2)**
>
> >**Q3**: Theory is not strong. The analysis is only conducted for strongly-convex-(strongly)-concave settings, while most practical examples work under the nonconvex setting. In addition, there are quite a few works on general federated nonconvex-concave/PL/strong-concave setting (see https://arxiv.org/abs/2302.04249 and related works therein). It would be great to discuss why only strongly-convex-(strongly)-concave settings are studied here. Again, the analysis is mainly developed for general case without taking the DRO min-max structure into account.
>
> **A3**: We would like to re-iterate our key contributions in this work. In the strongly-convex-(strongly)-concave (SC-SC or SC-C) setting, existing algorithms cannot match the convergence rate of the centralized one and achieve the accelerated convergence rates, and our work resolves this open problem. We believe such theoretical contributions are valuable additions to the existing federated min-max optimization literature.
>
> **Why study strongly-convex-(strongly)-concave settings?** As shown in previous work [YWK+2022], one can convexify the non-convex problem and solve the reformulated convex problem, which leads to improved performance compared to previous methods. Therefore, we believe solving the SC-SC and SC-C efficiently makes theoretical and empirical contributions to the literature.
>
> On the other hand, we strongly agree it would be interesting to investigate how to deal with the non-convex setting in future work, and it is out of the scope of this work. Please refer to **A2** for discussion on leveraging the special structure of DRO studied in our work.
>
> >**Q4**: Experiments can be made more convincing.
>
> **A4**: Thank you for your suggestion. We will provide additional experiments on SCAFF-PD in our camera-ready version.
>
> We thank you again for your thoughtful review and comments. Please let us know if you have other questions or comments.
>
> [YWK+2022] Yaodong Yu, Alexander Wei, Sai Praneeth Karimireddy, Yi Ma, and Michael Jordan. Tct: Convexifying federated learning using bootstrapped neural tangent kernels. Advances in Neural Information Processing Systems, 35:30882–30897, 2022.

---

### Official Review · Reviewer_2GJD · 2023-11-01

**Soundness:** 2 fair
**Presentation:** 2 fair
**Contribution:** 2 fair
**Rating:** 5
**Confidence:** 4

**Summary:**

This paper presents a communication efficient fair and robust federated learning algorithm. The communication efficiency is achieved by performing multiple updates at local agents before the central server performing the aggregation with a gradient extrapolation step that achieves the similar effect as Nesterov's acceleration. Fairness of the DRO problem is achieved by incorporating a set of weights to different agents that are subject to a constrained set and can be regularized through the design of the regularization function. Experiments on synthetic and real datasets to prove the communication efficiency and model performance are provided. Theoretical analysis in terms of convergence is also conducted.

**Strengths:**

The problem that is studied is of interest to the federated learning community. The developed algorithm also seems to be able to achieve the desired objective in terms of the experimental performance.

**Weaknesses:**

The technical proof part is not rigorous enough. More details will be provided below. There are a couple of typos and unclear definitions, will also be provided below.

Major comments:
1. What is $\bar{\tau}$ in Condition 5.1 and how do you set $\gamma_0$?
2. Lemma B.2 is wrong and the proof is also wrong, which leads to the soundness of Theorem B.6 and Theorem 5.1. Specifically,
to prove
\begin{equation}
t_r(\frac{1}{\tau_r}+ \mu_{\boldsymbol{x}}) \geq \frac{t_{r+1}}{\tau_{r+1}},
 \end{equation}
we should not start from rewritting this equation, which already assumes that this inequality is correct. The procedure I did is shown below:
\begin{equation}
t_r(\frac{1}{\tau_r}+ \mu_{\boldsymbol{x}}) =\frac{t_r}{\tau_r} (1+ \mu_{\boldsymbol{x}}{\tau_r}) = \frac{t_r}{\tau_r} \frac{\gamma_{r+1}}{\gamma_r}  = \frac{\sigma_r}{\sigma_0\tau_r} \frac{\gamma_{r+1}}{\gamma_r}  = \frac{\sigma_r}{\sigma_0\tau_r} \frac{\sigma_{r+1}/\tau_{r+1}}{\sigma_r/\tau_r}   = \frac{\sigma_{r+1}/\tau_{r+1}}{\sigma_0} =  \frac{t_{r+1}}{\tau_{r+1}}.
 \end{equation}
That means there is no greater than or equal to relationship, the two sides always equal to each other. Meanwhile, $\frac{t_r}{\sigma_r} = \frac{t_{r+1}}{\sigma_{r+1}}$.

3. From the assumptions and the definition of all symbols, I recognized that $\tau_r = J\eta_l\eta_g$, which indicates that $\tau_{r+1} = J\eta_l\eta_g$, too, if I did not miss anything. That means, for equation B.38, with $\frac{t_r}{t_r+1} = \theta_r$, we must have $V_{r+1}\leq \theta_{r+1}Z_{r+1}$, which however, is based on the condition that $\theta_{r+1}>1$. However, $\theta_{r+1} = \frac{\sigma_r}{\sigma_{r+1}} = \frac{\gamma_r\tau_r}{\gamma_{r+1}\tau_{r+1}} = \frac{\tau_r}{(1+ \mu_{\boldsymbol{x}}\tau_r)\tau_{r+1}} = \frac{1}{1+ \mu_{\boldsymbol{x}}{\tau_r}}<1$, which contradicts the assumption, which leads to me doubt about the soundness of the theorems. Please correct me if I am wrong.

4. The experiments on synthetic and real datasets seem contradict each other. For example, Figure 2 shows that larger $\rho$ leads to higher convergence rate while Figure 3 as well as in the analysis says “Meanwhile, the experimental results suggest that smaller $\rho$ leads to faster convergence w.r.t. worst-20% accuracy for our algorithm.” Which one is correct and why larger/smaller $\rho$ leads to faster convergence?

I am willing to discuss and change my rating if my comments can be addressed.

Minor comments:
1. Page 4, the line before equation (3.2), "iFor" ==> "For".
2. Page 5, "The extrapolation step used in Eq. (4.1) is to Nesterov’s acceleration (Nesterov, 2003)" ==> "The
extrapolation step used in Eq. (4.1) is $\textbf{similar}$ to Nesterov’s acceleration (Nesterov, 2003)".
3. Page 6, "We first introduce how to $\textbf{choice}$ the parameters for SCAFF-PD $\textbf{in when is}$ convex and {fi}i2[N]
are strongly convex in Condition 5.1.", some words seem missing.
4. First paragraph in Section 6, “After conducting thorough evaluations, we have observed that our proposed accelerated $\textbf{algorithms achieve}$ fast convergence rates and strong empirical performance on real-world datasets.” “After conducting thorough evaluations, we have observed that our proposed accelerated $\textbf{algorithm achieves}$ fast convergence rates and strong empirical performance on real-world datasets.”
5. Section 6.1, “we generate $y_i^i$ as $y^i_i = <\boldsymbol{a}_i^j, \hat{\boldsymbol{x}}+\delta_i^{\boldsymbol{x}}>$ “ ==>“we generate $y_i^i$ as $y^i_i = <\boldsymbol{a}_i^j, \hat{\boldsymbol{x}}>+\delta_i^{\boldsymbol{x}}$.
6. Page 8, “$\textbf{beside}$ the average classification accuracy across clients, we also evaluate the worst-20% accuracy1 for comparing fairness and robustness of different federated learning algorithms.” == > “$\textbf{besides}$ the average classification accuracy across clients, we also evaluate the worst-20% accuracy1 for comparing fairness and robustness of different federated learning algorithms.”
7. In table 3 in Appendix C.4, the leftmost column should be CIFAR-10 instead of CIFAR-100.

**Questions:**

1. In equation (3.1), what is the range of each $\lambda_i$? Is there any constraints on all $\lambda_i$'s, for example, does $\sum_i \lambda_i =1$?
2. What is the definition of $\Delta$ in equation (3.3)?
3. What is the definition of $D(\boldsymbol{\lambda}, \boldsymbol{\lambda}^r)$ in equation (4.2)?
4. Corollary 5.2, what is the definition of $\epsilon$?
5. What is the definition of $L_{xx}$ in equation (5.3)?
6. In the Introduction part, you use $T$ to indicates the convergence while in the theorems presented, you use $R$. Shouldn’t those be consistent?
7. For figure 2, how is the local update iteration $J$ chosen?

---

> ### Author Response · Authors · 2023-11-22
> **Author response (part 1)**
>
> >**Q1**: what is $\bar{\tau}$ in Condition 5.1 and how do you set $\gamma_0$.
>
> **A1**: We set $\bar{\tau}$ and $\gamma_0$ such that they satisfy the condition described in (B.35), for example, $\bar{\tau} = O(1/L_{xx})$ and $\sigma = O(L_{xx}/L^2_{\lambda x})$. $\bar{\tau}$ and $\gamma_0$ can be interpreted as the initial values for series {$\{\tau_r, \gamma_r\}$}.
>
> >**Q2**: Lemma B.2 is wrong and the proof is also wrong, which leads to the soundness of Theorem B.6 and Theorem 5.1... That means there is no greater than or equal to relationship, the two sides always equal to each other. Meanwhile, $\frac{t_r}{\sigma_r} = \frac{t_{r+1}}{\sigma_{r+1}}$.
>
> **A2**: We would like to clarify that, as you mentioned the two terms equal to each other, the ‘greater or equal to’ result still holds, therefore the argument and proof of Lemma B.2 is correct. We will revise the proof of Lemma B.2 to improve the presentation clarity.
>
> >**Q3**: From the assumptions and the definition of all symbols, I recognized that $\tau_r=J\eta_{\ell}\eta_{g}$, which indicates that ... which contradicts the assumption, which leads to me doubt about the soundness of the theorems. Please correct me if I am wrong.
>
> **A3**: We would like to clarify that $\tau_r$ is a parameter that may change over the communication round, which is defined in Lemma B.2. The local learning rate $\eta_{\ell}$ is a parameter that depends on $\tau_r$, where $\eta_{\ell}=\tau_r / (J \eta_g)$ (defined in Lemma A.3). $\eta_g$ is the global learning rate and $J$ is the number of local steps, and both $J$ and $\eta_{g}$ are constants. When $\tau_r$ changes over communication rounds, the value of $\eta_{\ell}$ will change accordingly. Therefore, $\tau_r/\tau_{r+1}\neq 1$ and $\frac{\tau_r}{(1+\mu_x \tau_r)\tau_{r+1}} \neq \frac{1}{1+\mu_x \tau_r}$, which means the last equality in **Q3** does not hold. Thus, the value of $\theta_{r+1}$ does not contradict with the assumption.
>
> **To summary, the proof of the Theorem is correct.** We will clarify the definition of the local step $\eta_{\ell}$ in our revision.
>
> >**Q4**: The experiments on synthetic and real datasets seem contradict each other. For example, Figure 2 shows that larger $\rho$ leads to higher convergence rate while Figure 3 as well as in the analysis says “Meanwhile, the experimental results suggest that smaller $\rho$ leads to faster convergence w.r.t. worst-20% accuracy for our algorithm.” Which one is correct and why larger/smaller $\rho$  leads to faster convergence?
>
> **A4**: There seems to be a misunderstanding. Figure 2 and Figure 3 are visualizing two different quantities:
> - **The y-axis Figure 2 represents the distance to the optimal solution.** For the experiments in Figure 2, we consider the synthetic data setup and study the convergence of different algorithms --- whether they converge to the unique optimal saddle point solution as well as the convergence rate of each algorithm.
> - **The y-axis Figure 3 represents training/test accuracy of the model.** For the experiments in Figure 3, we study the performance of different algorithms on real data, and we evaluate the training/test accuracy instead of the distance to the optimal solution.
>
> *Both experiments are correct.* We apologize for the confusion. Thank you for your suggestion, and we will provide the experimental results of the distance to optimal solutions for real data (the ones shown in Figure 3) in our camera-ready version.

---

> ### Author Response · Authors · 2023-11-23
> **Author response (part 2)**
>
> >**Q5**: In equation (3.1), what is the range of each $\lambda$? Is there any constraints on all $\lambda$'s, for example, does $\sum \lambda_i = 1$? What is the definition of $\Delta$ in equation (3.3)?
>
> **A5**: The $\lambda$ vector is constrained in the set $\Lambda$, for example, the $\Lambda=\Delta^{N}=${$\lambda:\sum_i \lambda_i = 1, 0\leq \lambda_i \leq 1$}. The $\lambda$ vector is always contained in the simplex $\Delta^{N}$, and our formulation is flexible and also allows adding additional constraints to $\lambda$, for example, refer to the constraints described in the equation before 'Finally, we can recover the Q-FL' sentence on Page 4, i.e., $\Lambda=$ {$\lambda \in \Delta, \lambda_i \leq 1/(\alpha N)$}.
>
> >**Q6**: What is the definition of $D(\lambda, \lambda_r)$ in equation (4.2)?
>
> **A6**: $D(\lambda, \lambda^{r}) = \| \lambda - \lambda^{r} \|^{2}$, which is described in Lemma A.4. We will highlight this in our notation part in the main body.
>
> >**Q7**: Corollary 5.2, what is the definition of $\varepsilon$?
>
> **A7**: $\varepsilon$ is the upper bound of the square of the distance between the $R$-th round iterate $x^{R}$ and the optimal solution $x^{\star}$, i.e., $\| x^{R} - x^{\star} \|^{2}\leq \varepsilon$.
>
> >**Q8**: What is the definition of $L_{xx}$ in equation (5.3)?
>
> **A8**: $L_{xx}$ is the smoothness parameter for the primal variable $x$, and the smoothness condition is defined in Eq (3.4). Thank you for pointing this out, we have added the $L_{xx}$ condition to Theorem 5.1 and Theorem 5.5 in our revision.
>
> >**Q9**: In the Introduction part, you use $T$ to indicates the convergence while in the theorems presented, you use $T$. Shouldn’t those be consistent?
>
> **A9**: Thank you for your suggestion. We used $T$ and $R$ to denote the number of communication rounds. We have changed $T$ to $R$ in our introduction section.
>
> >**Q10**: For figure 2, how is the local update iteration $J$ chosen?
>
> **A10**: In Figure 2, we chose $J=100$, which is described in Sec 6.1 – ‘For both algorithms, we set the number of local steps to be 100 and select the algorithm parameters through grid search’. We also studied the effect of local update iterations $J$ in Figure 5 (b) in the appendix.
>
> We thank you again for your thoughtful review and comments. We will fix the typos and improve the presentation based on your suggestions and comments in our camera-ready version. Please let us know if you have other questions or comments.

---

### Official Review · Reviewer_v329 · 2023-11-01

**Soundness:** 2 fair
**Presentation:** 3 good
**Contribution:** 2 fair
**Rating:** 5
**Confidence:** 4

**Summary:**

The authors introduce SCAFF-PD, which enhances fairness by optimizing distributionally robust objectives customized for diverse clients. They then employ an accelerated primal-dual (APD) algorithm with bias-corrected local steps, similar to SCAFFOLD, to improve communication efficiency and convergence speed.

**Strengths:**

1. This paper is easy to follow.
2. Building upon the foundation of SCAFFOLD, a new algorithm is developed for addressing distributionally robust federated objectives, and its convergence rate is rigorously derived.

**Weaknesses:**

1. The algorithm design and theoretical analysis rely on SCAFFOLD, encompassing the hypothesis and proof framework. This extensive reliance on prior work may potentially diminish the originality and contribution of the proposed method in this paper.
2. A notable issue arises in the algorithm design, as it necessitates two times of communications with nodes at each round, transmitting distinct content. This introduces a huge communication overhead. Additionally, contradictory to the federated context where nodes may join or leave the network at any time, the proposed algorithm must consistently maintain a stable participation at every round. This operational requirement may pose challenges to the practical applicability of the proposed algorithm.
3. The comparative analysis is limited by the inclusion of a small number of old methods.

**Questions:**

Please clarify the reason to claim that the proposed method is communication-efficient, while the algorithm introduces additional communication overhead.

---

> ### Author Response · Authors · 2023-11-22
> **Author response**
>
> >**Q1**: The algorithm design and theoretical analysis rely on SCAFFOLD, encompassing the hypothesis and proof framework. This extensive reliance on prior work may potentially diminish the originality and contribution of the proposed method in this paper.
>
> **A1**: Thank you for your suggestions on highlighting our technical contributions and the novelty of this work. We summarize the technical challenges as follows: the gradient correction used in SCAFFOLD only considers minimization problems; we need to design new gradient correction components in SCAFF-PD, which are **significantly different from the ones in SCAFFOLD, for solving the minimax problems in DRO**. The key novelty of our paper is to tackle the above technical challenges by (1). designing new algorithm components to perform gradient correction for min-max DRO problems; and (2). analyze and control the drift error on each device such that the proposed algorithm converges to the optimal solution. To this end, our analysis shows that SCAFF-PD enjoys faster convergence rates compared to existing algorithms for solving DRO problems in the federate learning literature.
>
> >**Q2**: A notable issue arises in the algorithm design, as it necessitates two times of communications with nodes at each round, transmitting distinct content. This introduces a huge communication overhead. Additionally, contradictory to the federated context where nodes may join or leave the network at any time, the proposed algorithm must consistently maintain a stable participation at every round. This operational requirement may pose challenges to the practical applicability of the proposed algorithm.
>
> **A2**: First of all, our goal of this work is to develop a faster federated optimization algorithm to reduce the number of communication rounds and leverage the local computational resource on each device. We agree that our algorithm requires two rounds of communication for updating the model parameters. However, compared to all existing algorithms in the literature, our proposed algorithms achieve the fastest convergence rate (w.r.t. communication) in terms of the condition numbers of the problem. Thus, when the condition number (i.e., $L^2_{\lambda x}/(\mu_{x}\mu_{\lambda})$) becomes larger, our algorithm achieves provably faster convergence than existing ones. Furthermore, through our experimental evaluations, we found *our algorithm achieves faster convergence performance on real-world tasks/datasets* **when using the same communication rounds as existing algorithms**.
>
> Regarding the setting where each node could join or leave the network at any time, this is definitely an interesting problem to investigate. However, this is out of the scope of this work. Even for the simpler case we studied in this work, previous existing algorithms cannot even match the convergence rate of the centralized algorithms, and our work resolves this open problem. It would be interesting to investigate how to deal with this challenging situation in future work.
>
> >**Q3**: The comparative analysis is limited by the inclusion of a small number of old methods.
>
> **A3**: Thank you for your suggestion. As also suggested by Reviewer W5dA, we will add the comparison between our algorithm and ProxSkip in our camera-ready version.
>
> We thank you again for your thoughtful review and comments. Please let us know if you have other questions or comments.

---

### Official Review · Reviewer_W5dA · 2023-11-01

**Soundness:** 2 fair
**Presentation:** 3 good
**Contribution:** 3 good
**Rating:** 6
**Confidence:** 4

**Summary:**

In this paper, the authors presented a new algorithm for federated learning in heterogeneous setup. The main idea of their approach is based on three things: distributionally robust objective problem (the appropriate reformulation of the problem), application of Prima-Dual Hybrid Gradient Method (PDHG) and Scaffold algorithm to make the method with local updates. The authors provide both convergence guarantee for strongly convex-concave and strongly convex-strongly concave cases. The experimental results show the effectiveness of the method.

**Strengths:**

1. The new method solves the DRO problem in saddle-point reformulation.
2. The combination of two technique tackle the issue related to data heterogeneity.
3. SCAFF-DP achieves better rates than previous methods and the experiments support this.

**Weaknesses:**

1. The first thing is related to the paragraph about choosing $\psi$ and $\Lambda$. From the convergence analysis, $\Lambda$ is a bounded set. However, there is no discussion about this in the main part.
2. In the main part there is no expression for local stepsize.
3. The formulation of Theorem 5.5 is not full. There is no word about the smoothness of function $f$.
4. It is good that the authors compare Proxskip and SCAFF-PD theoretically, however, there are a lot of new algorithms of 5th generation of local methods (see some of them here in the literature review https://arxiv.org/pdf/2302.09832.pdf). Better to say some words about them and compare. Another interesting thing is related to ProxSkip for variational inequalities (see https://openreview.net/forum?id=ct_s9E1saB1). There is no comparison between SCAFF-PD and this algorithm.
5. In the experiments, there is no comparison between the performances of SCAFF-PD and ProxSkip.

**Questions:**

1. Could you explain whether your proposed method has a speed up related to number of clients and number of local steps, which is observed for SCAFFOLD?
2. Please, add explanation for derivation of eq. (B.59) from eq.(B.57)

typos:
1.  In (A.1) one of $z$ in the last term have to be $x$.
2. in the first sentence of proof of Lemma A.4, probably, there is no need $4$ in denominator.
3. In the next sentence after eq. (B.17) in the formula for $\tau_r$, there is an extra bracket.

**Details Of Ethics Concerns:**

-

---

> ### Author Response · Authors · 2023-11-22
> **Author response (part 1)**
>
> >**Q1**: The first thing is related to the paragraph about choosing $\psi$ and $\Lambda$. From the convergence analysis, $\Lambda$ is a bounded set. However, there is no discussion about this in the main part.
>
> **A1**: Because in the dual update step, Eq. (4.1), the center (the *server*) is solving the constrained subproblem (in the second row of Eq. (4.1)), where $\lambda \in \Lambda$. In our proof, we applied the fact that $\lambda^{r+1}$ is the optimal solution for the constrained subproblem. Thank you for your suggestion, we will add more discussion about the constraint in the main part.
>
> >**Q2**: In the main part there is no expression for local stepsize.
>
> **A2**: The local step $\eta_{\ell}$ is defined in Lemma B.2. The local learning rate $\eta_{\ell}$ depends on $\tau_r$, where $\eta_{\ell}=\tau_r / (J \eta_g)$, where $\eta_g$ is the global learning rate and $J$ is the number of local steps, and both $J$ and $\eta_{g}$ are constants. We will clarify the definition of the local step $\eta_{\ell}$ in our revision.
>
> >**Q3**: The formulation of Theorem 5.5 is not full. There is no word about the smoothness of function $f$.
>
> **A3**: Thank you for your suggestion, we have included the smoothness conditions in Theorem 5.5 in our revision.
>
> >**Q4**: It is good that the authors compare Proxskip and SCAFF-PD theoretically, however, there are a lot of new algorithms of 5th generation of local methods (see some of them here in the literature review https://arxiv.org/pdf/2302.09832.pdf). Better to say some words about them and compare. Another interesting thing is related to ProxSkip for variational inequalities (see https://openreview.net/forum?id=ct_s9E1saB1). There is no comparison between SCAFF-PD and this algorithm.
>
> **A4**: Thank you for your suggestion. Our work focuses on solving the DRO problem (i.e., min-max optimization) problem, whereas most of the work described in the literature review [CAM+2023] are not designed for solving the DRO problem in federated learning. We will add related references from [CAM+2023] in our camera-ready version. Thank you for bringing this paper [ZL2022] to our attention. First of all, in terms of convergence rate, ProxSkip-VIP in [ZL2022] takes $\tilde{O}(\max(L_{xx}, L_{\lambda x})/\min(\mu_{x}, \mu_\lambda))$ communication rounds to converge to the $\varepsilon$-optimal solution, whereas our algorithm takes less number of communication rounds --- $\tilde{O}(L_{xx}/\mu_{x} + \sqrt{L_{\lambda x}^{2}/(\mu_x \mu_\lambda)})$, which is faster than ProxSkip-VIP. Secondly, it is not clear whether ProxSkip-VIP is guaranteed to converge in the strongly-convex-concave setting, and our algorithm converges to the optimal solution in this case. We will add the reference [ZL2022] as well as the above discussions to our revised version.
>
> >**Q5**: In the experiments, there is no comparison between the performances of SCAFF-PD and ProxSkip.
>
> **A5**: Thank you for your suggestion. We will include and discuss the comparison between SCAFF-PD and ProxSkip in the main body of our camera-ready version.
>
>
> [CAM+2023] TAMUNA: Doubly Accelerated Federated Learning with Local Training, Compression, and Partial Participation. Laurent Condat, Ivan Agarský, Grigory Malinovsky, Peter Richtárik. https://arxiv.org/abs/2302.09832.
>
> [ZL2022] ProxSkip for Stochastic Variational Inequalities: A Federated Learning Algorithm for Provable Communication Acceleration. Siqi Zhang, Nicolas Loizou, OPT 2022.

---

> > ### Comment · Reviewer_W5dA · 2023-11-23
> >
> > Thank you for your response! All my questions have been answered.

---

> ### Author Response · Authors · 2023-11-22
> **Author response (part 2)**
>
> > **Q6**: Could you explain whether your proposed method has a speed up related to the number of clients and number of local steps, which is observed for SCAFFOLD?
>
> **A6**: This is again a great observation. We answer this question in two parts:
> - **Speedup wrt clients $N$.** Our setting and proof allows any $\Lambda \subseteq \Delta^{N}$ i.e. any subset of the simplex. In particular, suppose we set $\Lambda = e_1$  i.e. all the weight is placed only on client 1 with $\lambda_1=1$ and $\lambda_j=0$ for all other clients $j\neq 1$. In this case, our DRO formulation (1.1) reduces to $\min_{x} f_1(x)$. This is effectively becomes a single client optimization, and we should not expect any speedup w.r.t. $N$. More generally, the DRO problem can potentially result in different clients being weighed drastically differently, with the worst case putting all weight on a single client. Thus, we cannot have a speedup wrt the number of clients $N$.
>
> - **Speedup wrt local iterations $J$.** Our result with some minor modifications does in fact show a speedup wrt the number of local iterations $J$, similar to SCAFFOLD. We had chosen to focus mainly on the dependence of $\varepsilon$ in our proofs and in comparison with prior works. This is because all prior work for the DRO problem only had a $1/\varepsilon^2$ rate, compared to our much faster $1/\varepsilon$ (matching SCAFFOLD). However, the reviewer is absolutely right that for a fine-grained comparison to SCAFFOLD, we need to focus on the constants $J$ and $N$ as well.
> To show a speedup wrt $J$, we can make one of the two following modifications:
>
>   - **Option (i)** The computation of $c_i = g_i(x)$ in SCAFF-PD needs to be done over a batch-size $J$ every round. This would increase the gradient complexity $\times 2$ but does not change the theoretical rates. However, this results in $c_i$ and $c$ having a much smaller variance of $\zeta^{2}/J$, and can be used to improve Lemma A.2 to derive $\mathcal{E}_{r} \leq \frac{12\tau^2}{\eta_g^2} \mathbb{E}[\| \nabla_x \Phi (x^{r}, \lambda^{r+1})\|^{2}] + \frac{15\tau^2}{\eta_g^2}(1+\chi)\frac{\zeta^{2}}{\color{red}J}.$ This improvement wrt $J$ can then be propagated in the rest of the proof where Lemma A.2 is used - e.g. in Eqn. (B.17). Doing this gives us the required speedup w.r.t. $J$ in all our results, including Corollary 5.6.
>
>   - **Option (ii)** Alternatively, we can use the gradients computed in the past round to compute . This is exactly the same as *option (ii) in the SCAFFOLD algorithm* --- see Eqn. (5) in [KKM+2020]. This would be more efficient since we do not need to compute additional stochastic gradients for $c_i$, however the proof is even more complex and messy.
>
> We chose the simpler approach in this work since the focus of our work was the min-max component of the DRO. We would be happy to extend our analysis for this version of SCAFF-PD if the reviewer thinks this would significantly improve our work.
>
> >**Q7**: Please, add explanation for derivation of eq. (B.59) from eq.(B.57)
>
> **A7**: We were using the fact that $\Delta_r \leq 0$ in the first inequality of Theorem B.9. We will add this explanation after Eq. (B.59) for better clarity.
>
> We thank you again for your thoughtful review and comments. We will fix the typos and improve the presentation based on your suggestions and comments in our camera-ready version. Please let us know if you have other questions or comments.
>
>
> [KKM+2020] SCAFFOLD: Stochastic Controlled Averaging for Federated Learning. Sai Praneeth Karimireddy, Satyen Kale, Mehryar Mohri, Sashank J. Reddi, Sebastian U. Stich, Ananda Theertha Suresh. ICML 2020.

---

> > ### Comment · Reviewer_W5dA · 2023-11-23
> >
> > Thank you for your response! All my questions have been answered.

---

### Meta-Review · Area_Chair_RwNc · 2023-12-09

**Metareview:**

The paper presents an algorithm called Scaff-PD designed for distributionally robust federated learning, aiming to improve fairness by tailoring distributionally robust objectives to heterogeneous clients. The resulting problem now becomes a min-max problem, where the max is take over weights within a prior constraint set and the min is taken over the model parameters to minimize the weighted loss across all clients.  A regularizer is added that aims to keep the weights not far from the average weights. The algorithm is an accelerated primal-dual (APD) approach that incorporates bias-corrected local steps inspired by Scaffold and an extrapolation step for faster convergence. In theory, the authors characterize the convergence rate for the proposed method for the strongly-convex-concave and strongly-convex-strongly-concave geometry, which matches with centralized version under certain conditions. Experiments are provided to demonstrate the effectiveness of the method.

Strengths of this paper:
The work is well written. Distributionally robust objectives in FL are important to achieve certain-level fairness by optimizing the worst-case distribution over clients rather than their simple average. The formulated problem is easy to following and well structured. The proposed algorithms incorporate several ideas in the primal-duel designs and the theory shows some improvements.

Weaknesses of the paper:
1) The formulation seems not to be quite new given existing studies. Although the authors show that their formulation is a generalization of existing ones, directly applying the distributionally robust objective (i.e., client weights optimized over a set of candidate distribution) is not new. The regularizer seems not to add too much new things into the formulation as well. In sum, although the formulation is general, the idea and individual components are not new.

2) The algorithms do not incorporate many new stuffs. For example, the SCAFFORD bias correction, the extrapolated step for min-max acceleration have been well studied.

3) Theory is not strong. The analysis is only conducted for strongly-convex-(strongly)-concave settings, while most practical examples work under the nonconvex setting.

4) Experiments can be further improved.

While the reviewers all agree the importance of the problem, the majority thinks the current version is weak and slightly below the acceptance borderline.

**Justification For Why Not Higher Score:**

While the reviewers all agree the importance of the problem, the majority thinks the current version is weak and slightly below the acceptance borderline. One reviewer weakly accepts it as slightly above the borderline after the rebuttal period.

**Justification For Why Not Lower Score:**

N.A.

---

### Decision · Program_Chairs · 2024-01-16

Reject